# Learning Exponential Families from Truncated Samples

**Jane H. Lee**
Department of Computer Science
Yale University
jane.h.lee@yale.edu

**Andre Wibisono**
Department of Computer Science
Yale University
andre.wibisono@yale.edu

**Manolis Zampetakis**
Department of Computer Science
Yale University
emmanouil.zampetakis@yale.edu

## Abstract

Missing data problems have many manifestations across many scientific fields. A fundamental type of missing data problem arises when samples are *truncated*, i.e., samples that lie in a subset of the support are not observed. Statistical estimation from truncated samples is a classical problem in statistics which dates back to Galton, Pearson, and Fisher. A recent line of work provides the first efficient estimation algorithms for the parameters of a Gaussian distribution [10] and for linear regression with Gaussian noise [11, 14, 37].

In this paper we generalize these results to log-concave exponential families. We provide an estimation algorithm that shows that *extrapolation* is possible for a much larger class of distributions while it maintains a polynomial sample and time complexity on average. Our algorithm is based on Projected Stochastic Gradient Descent and is not only applicable in a more general setting but is also simpler than the recent algorithms of [10, 26, 11, 14, 37]. Our work also has interesting implications for learning general log-concave distributions and sampling given only access to truncated data.

## 1 Introduction

In many statistical estimation and inference problems, we have access to only a limited part of the data that would be necessary for the classical statistical methods to work, which motivates the development of statistical methods that are resilient to *missing data* [29]. *Truncation* [32, 8] is a fundamental and frequent type of missing data and arises when samples that lie outside a subset of the support are not observed and their count is also not observed. Statistical estimation from truncated samples is the focus of the field of truncated statistics, which was developed since the beginning of the twentieth century starting with the work of Galton [19], Pearson and Lee [34, 35], and Fisher [16]. Truncated statistics is widely applicable in Econometrics and many other theoretical and applied fields [32].

A recent line of work establishes the first sample optimal and computationally efficient methods for fundamental statistical estimation problems from truncated samples [10, 26, 11, 13, 14, 24, 37]. All the aforementioned works though heavily rely on the Gaussianity of the distribution of data or the Gaussianity of the noise in regression problems. Gaussianity is an idealized assumption and the question of generalizing truncated statistics beyond Gaussianity has been explored in many existing works, e.g., [1, 22, 38]. The only results in this regime though are for single dimensional problems and truncations that can be described as intervals.

37th Conference on Neural Information Processing Systems (NeurIPS 2023).

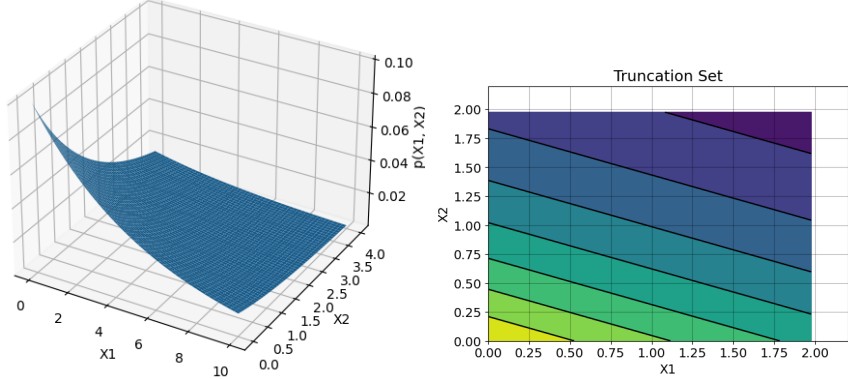

Figure 1: Visualizing the density and truncation set for 2-dimensional exponential distribution. The plot on the left is the density of $p(x_1, x_2) \propto \exp(-\frac{x_1}{5} - \frac{x_2}{2})$. On the right is a contour plot of the truncation set $[0, 2] \times [0, 2]$ under $p(x_1, x_2)$. Note that $\mathbb{E}[X_1] = 5$ and $\mathbb{E}[X_2] = 2$, so the truncation set excludes the mean in one direction and includes it only as a boundary point in the other.

In this work we provide *statistically and computationally efficient methods for estimating the parameters of exponential families from truncated samples*. Our results generalize the recent work of [10] and are the first to provide an estimation algorithm for this problem for a general class of exponential families and for a general class of truncation biases.

*Exponential families* are one of the most influential type of distribution classes since they include many fundamental distributions such as normal, exponential, beta, gamma, chi-squared, and Weibull distributions. They were first introduced by Fisher [17] and later generalized by [9, 27, 36]. The estimation of the parameters of exponential families over continuous domains is the subject of many classical and recent results; starting from the work of Fisher [17] until the recent results of [25, 41]. This line of work has also found applications in many areas of statistics including causal inference [40]. Our work contributes in this line of research as well since we show how to estimate exponential families even when we only have access to truncated samples.

## 1.1  Our Results

For every distribution $p$, we define the truncated version $p^S$ as the distribution with density that satisfies $p^S(\mathbf{x}) \propto \mathbf{1}\{\mathbf{x} \in S\} \cdot p(\mathbf{x})$. We consider distributions $p_{\boldsymbol{\theta}}$ with support $\mathcal{X} \subseteq \mathbb{R}^m$ parameterized by a vector of parameters $\boldsymbol{\theta} \in \mathbb{R}^k$ that can be written in the following form:

$$p_{\boldsymbol{\theta}}(\mathbf{x}) \propto h(\mathbf{x}) \cdot \exp(\boldsymbol{\theta}^\top T(\mathbf{x}))$$

where $h$ and $T$ are known functions and $T$ is called the *sufficient statistics* of the exponential family. Our goal is to estimate the vector of parameters $\boldsymbol{\theta}$ up to $\epsilon$ error in $\ell_2$ distance using only samples from $p_{\boldsymbol{\theta}}^S$ when we have oracle access to $S$, i.e., there is an oracle that for every $\mathbf{x} \in \mathcal{X}$ can answer whether $\mathbf{x} \in S$ or not.

Without any further assumptions estimation from truncated samples is impossible as shown in [10]. In this paper we use the following assumptions to get our estimation guarantees:

**Assumption I:** We assume a lower bound and an upper bound on the variance of the sufficient statistics in any direction. This assumption is used in estimation of the parameters of exponential families even without truncated samples, e.g., Assumption 4.1 in [41].

**Assumption II:** We assume that the exponential families contain log-concave distributions. We use this assumption in two places: (1) to show our extrapolation result, our distributions must satisfy an anti-concentration property; this is a property that is heavily utilized even in estimating a Gaussian distribution [10], (2) we require access to a sampling oracle for the underlying distributions; this sampling oracle can be implemented efficiently using Langevin dynamics if we have log-concavity.

**Assumption III:** We assume that the sufficient statistics $T$ are bounded degree polynomials. This assumption combined with log-concavity provides the anti-concentration property that we

need. Alternatively, we can assume that the sufficient statistics belongs to a function class that satisfies anti-concentration but for simplicity of exposition we focus on the case where $T(\mathbf{x})$ is a polynomial.

Our main result is the following.

**Informal Theorem 1.1** (See Theorem 3.1 for the formal statement)**.** Under Assumptions I, II, III, and given samples $n$ (at least $\widetilde{\mathcal{O}}(k/\epsilon^2)$ for some $\epsilon$) from $p_{\boldsymbol{\theta}*}^S$, where $S$ is a measurable set to which we have oracle access, there exists an estimation algorithm with (expected) running time $\mathrm{poly}(m, k, n)$ that outputs $\widehat{\boldsymbol{\theta}}$ such that with probability at least 99%, it holds that $\|\widehat{\boldsymbol{\theta}} - \boldsymbol{\theta}^*\| < \epsilon$.

Our main result has the following important implications:

▷ We show that our assumptions are satisfied from exponential distributions, Weibull distributions, continuous Bernoulli, continuous Poisson, Gaussian distributions, and generalized linear models. Hence, our result implies an efficient method for estimation from truncated samples for all these distribution classes.

▷ Another interesting corollary of our result is that we can combine it with the ideas of [12] and get a general method for learning log-concave distributions from truncated samples. In particular, assume that we want to learn from truncated samples a distribution that can be written in the form $p(\mathbf{x}) \propto \exp(-f(\mathbf{x}))$ where $f$ is a convex function. Now under mild assumptions, we can replace $f(\mathbf{x})$ with a finite Taylor approximation, i.e., we have that $f(\mathbf{x}) \cong \sum_i a_i t_i(\mathbf{x})$ for some polynomials $t_i(\mathbf{x})$. Then, using our method we can estimate the parameters $a_i$ and output an estimation of $p$.

▷ In the context of sampling, the gradient of the log-likelihood is the score function that is needed to run Langevin dynamics (e.g., see [30, 7]). Our result also says that we are able to sample from the original distribution, given that we only observe truncated samples.

**Technical Contributions.** Our algorithm is *projected stochastic gradient descent (PSGD)* on the *negative log-likelihood* function and is the same algorithm that is used in many of the recent works in truncated statistics. As in the previous work though, the main challenge is to show that PSGD will converge in our general setting. The previous analysis of the convergence of PSGD was based on exact properties of the Gaussian distribution. When we move away from Gaussianity every step of the convergence analysis becomes much more technical and we need to make sure to only use properties that generalize to exponential families beyond Gaussian distributions. In particular, certain results such as that of Lemma 3.6 and its related quantities in C.1 can be generalized even beyond exponential families, holding for any density.

## 1.2 Related Work

Our most related literature is the recent series of works on truncated statistics which includes the following results: estimation of multivariate normal distributions [10], linear regression with Gaussian noise [10, 26, 11, 13, 14, 37], estimation of product distributions over the hypercube [18], non-parametric density estimation [12]. All of these works heavily rely on properties of the Gaussian distributions, or product distributions over the hypercube, or their dependence in the number of dimensions is not efficient, e.g., [12]. In our work we identify the properties of exponential families that are only required to get the efficient estimation results and we show that linear dependence on the dimension is achievable in settings that are more general than the Gaussian case.

Another related work is that of [30] that solves parameter estimation of a truncated density given samples through the score matching technique. To derive a tractable objective, we need appropriate boundary conditions which are not satisfied by truncated densities, but [30] instead uses a modified weighted Fisher distance given that the truncation set $S$ is a Lipschitz domain (a type of open and connected set). On the other hand, our work assumes no particular structure about $S$ and hence our results are more general and applicable in much more complicated settings for exponential families.

## 2 Preliminaries

**Notation.** Lowercase bold letters will denote real-valued vectors, e.g., $\mathbf{x} \in \mathbb{R}^m$, and uppercase bold letters will denote matrices with real values, e.g., $\mathbf{A} \in \mathbb{R}^{n \times m}$. For a random vector $\mathbf{x} \sim \rho$,

$\mathbf{Cov}[\mathbf{x}] = \mathbf{Cov}[\mathbf{x}, \mathbf{x}] = \mathbb{E}[(\mathbf{x} - \mathbb{E}[\mathbf{x}])(\mathbf{x} - \mathbb{E}[\mathbf{x}])^\top]$ is its covariance matrix, and $\mathbf{Var}(\mathbf{x})$ is the trace of the covariance matrix (a scalar value). Depending on whether it is clear from context, $\mathbf{Cov}$ and $\mathbf{Var}$ may include subscripts to indicate the distribution $\rho$. The notation $B(c, R)$ is the Euclidean ball centered at $c$ with radius $R > 0$.

**Exponential Families.** Let $\mathbf{x} \in \mathcal{X} \subseteq \mathbb{R}^m$. We are interested in a class of densities which have the form,
$$p_\theta(\mathbf{x}) = h(\mathbf{x}) \exp(\boldsymbol{\theta}^\top T(\mathbf{x}) - A(\boldsymbol{\theta})),$$
where $h : \mathbb{R}^m \mapsto \mathbb{R}_+$ is the *base* or *carrier measure*, $\boldsymbol{\theta} \in \Theta$ with $\Theta = \{\boldsymbol{\theta} \in \mathbb{R}^k : A(\boldsymbol{\theta}) < \infty\}$ is the *natural parameter space*, $T : \mathbb{R}^m \mapsto \mathbb{R}^k$ is the *sufficient statistic* for $\boldsymbol{\theta}$, and $A(\boldsymbol{\theta}) = \log Z(\boldsymbol{\theta})$ is the log-partition function, where $Z(\boldsymbol{\theta}) = \int p_\theta(\mathbf{x}) d\mathbf{x}$.

A *regular* exponential family is one where $\Theta$ is an open set. It is *minimal* if the $\boldsymbol{\theta}$ and $T(\mathbf{x})$ are each linearly independent. Any non-minimal family can be made minimal by appropriate reparametrization. In any regular exponential family, $A(\boldsymbol{\theta})$ is convex. It is strictly convex if the representation is minimal. Exponential families have several nice properties (e.g., see Theorem 1 of [4]), among which are that $\nabla A(\boldsymbol{\theta}) = \mathbb{E}_{p_\theta}[T(\mathbf{x})]$ and $\nabla^2 A(\boldsymbol{\theta}) = \mathbf{Cov}_{p_\theta}[T(\mathbf{x})]$.

**Truncated Distributions.** Let $\rho$ be a probability distribution on $\mathbb{R}^m$. We represent $\rho$ as a probability density function with respect to the Lebesgue measure $d\mathbf{x}$ on $\mathbb{R}^m$. Let $S \subseteq \mathbb{R}^m$ be such that $\rho(S) = \alpha$ for some $\alpha \in (0, 1]$. Let $\rho^S := \rho(\cdot \mid \cdot \in S)$ be the conditional distribution of $\mathbf{x} \sim \rho$ given that $\mathbf{x} \in S$. Concretely, the density of $\rho^S$ is
$$\rho^S(\mathbf{x}) = \frac{\rho(\mathbf{x}) \cdot \mathbb{1}\{\mathbf{x} \in S\}}{\rho(S)}.$$
For exponential families, we have the truncated density $p_\theta^S(\mathbf{x})$ is:
$$p_\theta^S(\mathbf{x}) = \frac{p_\theta(\mathbf{x})}{\int_S p_\theta(\mathbf{x}) d\mathbf{x}} \mathbb{1}\{\mathbf{x} \in S\} = \frac{h(\mathbf{x}) \exp(\boldsymbol{\theta}^\top T(\mathbf{x}))}{\int_S h(\mathbf{x}) \exp(\boldsymbol{\theta}^\top T(\mathbf{x})) d\mathbf{x}} \mathbb{1}\{\mathbf{x} \in S\}.$$
See Figure 1 for an illustration.

**Sub-Exponential Distributions.** Although the term sub-exponential has been overloaded (e.g., [21] v.s.[44]), the definition we will use describes a class of distributions whose tails decay at least as fast as an exponential, but with potentially heavier tails than Gaussians [44].

There are several equivalent characterizations of sub-exponential random variables (e.g., see Prop. 2.7.1 of [44]), one of which uses the moment generating function.

**Definition** (Sub-exponential random variable). A centered, real-valued random variable $X \in SE(\nu^2, \beta)$ is sub-exponential with parameters $\nu^2, \beta > 0$ if
$$\mathbb{E}[e^{\lambda X}] \leq e^{\frac{\nu^2 \lambda^2}{2}}, \quad \forall \lambda : |\lambda| < 1/\beta.$$

**Membership Oracle of a Set.** Let $S \subseteq \mathbb{R}^m$. A *membership oracle* is an efficient procedure which computes $\mathbb{1}\{\mathbf{x} \in S\}$.

## 3 Projected Stochastic Gradient Descent Algorithm

**Problem Setup.** We are given truncated samples $\{\mathbf{x}_i\}_{i=1}^n$, with each $\mathbf{x}_i \sim p_{\boldsymbol{\theta}^*}^S$, where $p_{\boldsymbol{\theta}^*}(S) = \alpha > 0$. Without knowledge of the truncation set $S$ beyond access to a membership oracle, can one recover $\boldsymbol{\theta}^*$ and thus $p_{\boldsymbol{\theta}^*}$ efficiently?

We answer this question positively, under the following assumptions:

**Assumption A1** (Strong Convexity, Smoothness of Non-truncated Negative Log-Likelihood over $\Theta$).
$$\lambda I \preceq \mathbf{Cov}_{\mathbf{z} \sim p_\theta}[T(\mathbf{z}), T(\mathbf{z})] \preceq LI \qquad \forall \boldsymbol{\theta} \in \Theta,$$
for some $\lambda, L > 0$. Here, we've abused notation for $\Theta$ which can be a subset of the entire natural parameter space. As mentioned earlier, this is always at least convex for exponential families and strictly convex in minimal representation. Thus the negative log-likelihood (of the non-truncated density) can be made strongly convex and smooth by restricting the natural parameter space appropriately.

**Assumption A2** (Log-Concave Density). The density $p_{\boldsymbol{\theta}}(\mathbf{x})$ is log-concave in $\mathbf{x}$.

**Assumption A3** (Sufficient Statistics $T(\mathbf{x})$ is polynomial in $\mathbf{x}$). $T(\mathbf{x}) \in \mathbb{R}^k$ has components which are polynomial in $\mathbf{x}$, with degree at most $d$.

Assumptions A2 and A3 allow us to use the anti-concentration result needed for Lemma 3.2 which is heavily utilized even in the Gaussian case. While A2 also allows for efficient sampling via Langevin dynamics, the latter is only used in Lemma 3.2. Refer back to 1.1 for discussion of these assumptions.

**Main Result.**

**Theorem 3.1** (Main). Given membership oracle access to a measurable set $S$ whose measure is some constant $\alpha \in (0, 1]$ under an unknown exponential family distribution $p_{\boldsymbol{\theta}^*}$ which satisfies A1, A2, A3, and given samples $\mathbf{x}_1, \ldots, \mathbf{x}_n$ from $p_{\boldsymbol{\theta}^*}$ that are truncated to this set, there exists an expected polynomial-time algorithm that recovers an estimate $\widehat{\boldsymbol{\theta}}$. That is, for any $\epsilon > 0$ the algorithm

- Uses an expected $\widetilde{\mathcal{O}}(k/\epsilon^2)$ truncated samples and queries to the membership oracle,

- Runs in expected $\mathrm{poly}(m, k, 1/\epsilon)$ time.

- Produces an estimate $\widehat{\boldsymbol{\theta}}$ such that with probability at least 99%,

$$\|\widehat{\boldsymbol{\theta}} - \boldsymbol{\theta}^*\| < \epsilon.$$

In order the solve this problem, we need to define an objective whose optimum is $\boldsymbol{\theta}^*$ and we need to be able to recover it uniquely. To use maximum likelihood estimation (or minimize the negative log-likelihood), we have to be able to compute gradients which depend on the truncation set $S$, which we cannot do directly without more knowledge about $S$. However, we can sample unbiased estimates of the gradient, as long we have non-trivial mass on $S$ at a current parameter estimate (otherwise the truncated likelihood function at that parameter is not well-defined and rejection sampling would take infinite time). To address all of these issues, the organization of this section is as follows:

- Section 3.1 establishes that after truncation, the negative log-likelihood remains strongly convex and smooth (in $\boldsymbol{\theta}$) over a subset of parameters which have non-trivial mass on the truncation set.

- In Section 3.3, we show that while we do not know the truncation set, we can solve the non-truncated MLE problem with truncated samples to find an initial parameter $\boldsymbol{\theta}_0$ which assigns non-trivial mass to the truncation set.

- Then given this $\boldsymbol{\theta}_0$, in Section 3.2 we show that we can construct a set of parameters $K$ which all assign non-trivial mass to the truncation set (and contains the true parameter $\boldsymbol{\theta}^*$).

- In Section 3.4, we use results from the previous sections to prove that we can efficiently recover the true parameter $\boldsymbol{\theta}^*$ using a stochastic gradient descent procedure minimizing the truncated negative log-likelihood, which projects to the parameter space $K$.

## 3.1 Strong Convexity and Smoothness of Truncated Negative Log-Likelihood

Without truncation, recovering the true parameter $\boldsymbol{\theta}^*$ for any parameterized distribution given samples is a classical problem solved by maximizing the likelihood (or minimizing its negation). Here, we state the main objective we will minimize through a stochastic gradient descent procedure as well as the properties of this objective that will allow us to recover $\boldsymbol{\theta}^*$. Define:

$$\bar{\ell}(\boldsymbol{\theta}) := -\mathbb{E}_{\mathbf{x} \sim p_{\boldsymbol{\theta}^*}^S} \left[ \log \left( p_{\boldsymbol{\theta}}^S(\mathbf{x}) \right) \right]$$

$$\nabla_{\boldsymbol{\theta}} \bar{\ell}(\boldsymbol{\theta}) = \mathbb{E}_{\mathbf{z} \sim p_{\boldsymbol{\theta}}^S}[T(\mathbf{z})] - \mathbb{E}_{\mathbf{x} \sim p_{\boldsymbol{\theta}^*}^S}[T(\mathbf{x})]$$

$$\nabla_{\boldsymbol{\theta}}^2 \bar{\ell}(\boldsymbol{\theta}) = \mathbf{Cov}_{\mathbf{z} \sim p_{\boldsymbol{\theta}}^S}[T(\mathbf{z}), T(\mathbf{z})]$$

Note that since the Hessian is a covariance matrix which is at least PSD, this objective is always convex in $\boldsymbol{\theta}$. Thus $\boldsymbol{\theta}^*$ is a minimizer since it satisfies the first-order optimality condition. (These calculations can be found in Appendix A.1.) However, if the objective is too flat, we may not be able to recover $\boldsymbol{\theta}^*$ even after sufficiently reducing the objective value. For this, we prove that if the original non-truncated covariance has bounded eigenvalues, the truncated one does as well under A1, A2, and A3 at parameters which assign non-trivial mass to $S$.

**Lemma 3.2** (Preservation of Strong Convexity under Truncation). Assume the lower bound in A1, A2, A3. If $p_{\boldsymbol{\theta}}(S) > 0$, then

$$\mathbf{Cov}_{\mathbf{z} \sim p_{\boldsymbol{\theta}}^S}[T(\mathbf{z}), T(\mathbf{z})] \succeq \frac{1}{2}\left(\frac{p_{\boldsymbol{\theta}}(S)}{4Cd}\right)^{2d} \lambda I,$$

where $C$ is a universal constant guaranteed by Theorem 8 of [5] and $d$ is the maximum degree of $T(\mathbf{x})$. See proof in Appendix A.2 which follows that of [10].

**Lemma 3.3** (Preservation of Smoothness under Truncation). Assume the upper bound in A1. Suppose $p_{\boldsymbol{\theta}}(S) > 0$, then

$$\mathbf{Cov}_{\mathbf{z} \sim p_{\boldsymbol{\theta}}^S}[T(\mathbf{z}), T(\mathbf{z})] \preceq \frac{1}{p_{\boldsymbol{\theta}}(S)}LI.$$

See proof in Appendix A.3. The proof is simple and can be done similarly to the previous lemma.

Thus, we have shown that as long as we optimize over a parameter space where every $\boldsymbol{\theta}$ assigns non-trivial mass to the truncation set, our objective is both strongly convex and smooth. The following sections will help us determine and then construct this set given samples.

*Remark.* Note that the upper bound increased and the lower bound decreased for the eigenvalues of the truncated covariance matrix. Even in a one-dimensional simple case like $\mathcal{N}(0, 1)$, it is easy to construct examples of both increasing the shrinking the variance given freedom to place mass anywhere under $\mathcal{N}(0, 1)$. Thus, it may be natural that the eigenvalue range expands after truncation.

*Remark.* Lemma 3.2 and the log-likelihood calculations are direct generalizations of prior work in the Gaussian case, where we can recover the results of [10] by noting that the re-parameterization of Gaussian parameters $(\boldsymbol{\mu}, \boldsymbol{\Sigma})$ as $\boldsymbol{\nu} = \boldsymbol{\Sigma}^{-1}\boldsymbol{\mu}$ and $\mathbf{T} = \boldsymbol{\Sigma}^{-1}$ is the natural parameterization of multivariate Gaussian distributions in exponential family form (up to constants). The sufficient statistics here $T(\mathbf{x}) \propto [\mathbf{x}, \mathbf{x}\mathbf{x}^\top]$ has components which are polynomial in $\mathbf{x}$ with degree at most 2, and plugging in $d = 2$ to Lemma 3.2 recovers Lemma 4 in [10]. Appendix B includes other examples beyond Gaussians which satisfy A1, A2, and A3.

## 3.2 Parameter Space with Non-Trivial Mass on Truncation Set

The prior section established that the strong convexity and smoothness parameter of the truncated objective is controlled by the mass that $p_{\boldsymbol{\theta}}$ assigns to the truncation set $S$ for any given $\boldsymbol{\theta}$. In this section, we will prove lower bounds on the mass that $p_{\boldsymbol{\theta}}$ assigns to the truncation set, given that the parameter distance to the optimum $\|\boldsymbol{\theta} - \boldsymbol{\theta}^*\|$ is bounded.

**Lemma 3.4** (Lower bound for mass on truncation set under smoothness given parameter distance). Assume A1. Let $\boldsymbol{\theta}, \boldsymbol{\theta}' \in \Theta$. Then for two distributions from the same exponential family

$$p_{\boldsymbol{\theta}}(S) \geq p_{\boldsymbol{\theta}'}(S)^2 \cdot \exp\left(-\frac{3L}{2}\|\boldsymbol{\theta} - \boldsymbol{\theta}'\|^2\right).$$

Proof is provided in Appendix C.3, and only needs smoothness. Thus, we can lower bound the mass that a parameter $\boldsymbol{\theta}$ assigns to $S$ given its distance $\|\boldsymbol{\theta} - \boldsymbol{\theta}^*\|$ from $\boldsymbol{\theta}^*$ which is assumed to have $p_{\boldsymbol{\theta}^*}(S) = \alpha$.

Thus, to make use of this property we want to establish a procedure to initialize a parameter $\boldsymbol{\theta}_0$ such that its distance to the optimum $\boldsymbol{\theta}^*$ is bounded. Then during the optimization procedure we will make sure to make progress toward $\boldsymbol{\theta}^*$. The following is the result we will be able to prove after proving some results between truncated and non-truncated quantities in the following Section 3.3.

**Corollary 3.5** (Parameter space with non-trivial mass on truncation set). Given $\boldsymbol{\theta}_0$ such that $E_{p_{\boldsymbol{\theta}_0}}[T(\mathbf{x})] = \overline{T}$ where $\overline{T} = \frac{1}{n}\sum_{i=1}^n T(\mathbf{x}_i)$ is the empirical mean sufficient statistics given our samples $\{\mathbf{x}_i\}_{i=1}^n$ for each $\mathbf{x}_i \sim p_{\boldsymbol{\theta}^*}^S$, if we define

$$K = B\left(\boldsymbol{\theta}_0, \frac{\epsilon_S + \mathcal{O}(\log 1/\alpha)}{\lambda}\right) \cap \Theta$$

then $p_{\boldsymbol{\theta}}(S) \geq \alpha^2 \exp\left(-\frac{6L}{\lambda^2}(\epsilon_S + \mathcal{O}(\log 1/\alpha))^2\right) > 0$ holds $\forall \boldsymbol{\theta} \in K$ (with probability at least $1 - \delta$ for $n \geq \Omega(\log(1/\delta))$). Furthermore, $\boldsymbol{\theta}^* \in K$ (as long as $\boldsymbol{\theta}^* \in \Theta$ satisfies conditions of Claim 1).

This result will follow from Lemma 3.4, Claim 1, Lemma 3.7, and Lemma 3.9 in the next section.

## 3.3 Initialization with Empirical Samples and Non-truncated MLE

Given samples from the truncated density $p_{\boldsymbol{\theta}^*}^S$, one may first try to solve the non-truncated empirical MLE problem to find a parameter $\boldsymbol{\theta}_0$ without truncation and hope that it is good enough. (We will show that it is.) In order to understand how good this initial guess is, we need to establish some relationships between the truncated and non-truncated density.

**Lemma 3.6** (Truncated vs. Non-truncated Mean Sufficient Statistics for General Densities). Let $\rho$ be a probability distribution on $\mathbb{R}^d$ (not necessarily from an exponential family). Let $S \subseteq \mathbb{R}^d$ with $\rho(S) > 0$. Then

$$\|\mathbb{E}_{\rho^S}[\mathbf{x}] - \mathbb{E}_\rho[\mathbf{x}]\| \leq \sqrt{\frac{1 - \rho(S)}{\rho(S)}} \cdot \sqrt{\mathbf{Var}_\rho(\mathbf{x})}.$$

Proof of this lemma and several related quantities for general truncated densities is in Appendix C.1. In low dimensions, this variance term may effectively be a constant; however, in high-dimensional settings this term can grow with dimension (which is undesirable if we want an efficient algorithm). Given more assumptions about the density, we can get better dimension-free bounds which generalize the results from the Gaussian case.

**Claim 1.** Let $\boldsymbol{\theta} \in \Theta$ such that $\boldsymbol{\theta} + \frac{1}{\beta}\mathbf{u} \in \Theta$ for some $\beta > 0$ for all unit vectors $\mathbf{u}$ and such that Assumption A1 holds for $p_{\boldsymbol{\theta}}$ from an exponential family. Then $X := \mathbf{u}^\top (T(\mathbf{x}) - \mathbb{E}_{\mathbf{x} \sim p_{\boldsymbol{\theta}}}[T(\mathbf{x})])$ is $SE(L, \beta)$. (Proof provided in Appendix C.2.)

*Remark.* This claim allows us to have concentration of the empirical mean sufficient statistics to its population mean for the non-truncated distribution. The prior work [10] analyzed the mean and covariance of a Gaussian separately, but for instance to establish bounds on distances between the truncated and non-truncated mean parameter, it made use of Gaussian concentration inequalities (which are tighter than sub-exponential ones). The relationship between the truncated density and the non-truncated one will allow us to say that the empirical truncated mean sufficient statistics is also somewhat close to the non-truncated population mean.

**Lemma 3.7** (Concentration of Empirical vs. Non-truncated Mean Sufficient Statistics). Suppose $\boldsymbol{\theta}^*$ satisfies the conditions of Claim 1 and $p_{\boldsymbol{\theta}^*}(S) = \alpha > 0$. Let $\overline{T} = \frac{1}{n}\sum_{i=1}^n T(\mathbf{x}_i)$ be the empirical mean sufficient statistics given our samples $\{\mathbf{x}_i\}_{i=1}^n$ each $\mathbf{x}_i \sim p_{\boldsymbol{\theta}^*}^S$. Let $\epsilon_S > 0$. For $n \geq \Omega\left(\frac{2\beta}{\epsilon_S}\log\left(\frac{1}{\delta}\right)\right)$, with probability at least $1 - \delta$,

$$\|\overline{T} - \mathbb{E}_{p_{\boldsymbol{\theta}^*}}[T(\mathbf{x})]\| \leq \epsilon_S + \mathcal{O}(\log 1/\alpha).$$

It should be noted that it suffices to consider $\epsilon_S$ as some independent constant (and not related to the $\epsilon$ accuracy parameter in the main theorem). However, by taking $n$ at least $\widetilde{\mathcal{O}}(k/\epsilon^2)$ as stated in the main theorem, this will be small. See proof in Appendix C.4.

*Remark.* At a high level, the truncated samples can be thought of as $\mathcal{O}(n/\alpha)$ samples from the non-truncated distribution (keeping only those in $S$), and each are "not too far" (depending on how much mass the set $S$ has under the non-truncated distribution) from the non-truncated mean due to concentration. Note that the $\mathcal{O}(\log 1/\alpha)$ term will never disappear even as we increase $n$, which quantifies the inherent bias that the truncated mean sufficient statistics will have with respect to the non-truncated one (and is large if the mass $\alpha$ is small). From this, we can also say something about the population mean sufficient statistics, one on the truncated distribution and one on the non-truncated.

**Corollary 3.8** (Truncated vs. Non-truncated Mean Sufficient Statistics). Let $\boldsymbol{\theta}$ satisfy the conditions of Claim 1 and $p_{\boldsymbol{\theta}}(S) > 0$. Then

$$\|\mathbb{E}_{p_{\boldsymbol{\theta}}^S}[T(\mathbf{x})] - \mathbb{E}_{\mathbf{x} \sim p_{\boldsymbol{\theta}}}[T(\mathbf{x})]\| \leq \mathcal{O}(\log 1/p_{\boldsymbol{\theta}}(S)).$$

The proof follows from the preceding lemma, replacing $\alpha$ with $p_{\boldsymbol{\theta}}(S)$ and taking $n \to \infty$. Compare this to the Gaussian case [10], where the mean and truncated means were bounded as $\|\boldsymbol{\mu} - \boldsymbol{\mu}_S\| \leq \mathcal{O}(\sqrt{\log 1/p_{\boldsymbol{\theta}}(S)})$ and separately the covariances were bounded as $\|\boldsymbol{\Sigma}^{-1/2}\boldsymbol{\Sigma}_S\boldsymbol{\Sigma}^{-1/2} - \mathbf{I}\|_F \leq \mathcal{O}(\log 1/p_{\boldsymbol{\theta}}(S))$. Note the smaller $\mathcal{O}(\sqrt{\log 1/p_{\boldsymbol{\theta}}(S)})$ quantity due to tighter Gaussian concentration vs. the sub-exponential rate.

Once we have bounds on the norm of the difference between the truncated and non-truncated mean sufficient statistics, we can bound distance in parameter space. The following completes this.

**Lemma 3.9** (Non-truncated MLE Solution Distance to $\boldsymbol{\theta}^*$). Suppose $\boldsymbol{\theta}^*$ satisfies the conditions of Claim 1 and $p_{\boldsymbol{\theta}^*}(S) = \alpha > 0$. Let $\boldsymbol{\theta}_0$ be such that $\mathbb{E}_{p_{\boldsymbol{\theta}_0}}[T(\mathbf{x})] = \overline{T}$ where $\overline{T} = \frac{1}{n}\sum_{i=1}^{n} T(\mathbf{x}_i)$ given each $\mathbf{x}_i \sim p_{\boldsymbol{\theta}^*}^S$. Let $\epsilon_S > 0$ and $n > \Omega\left(\frac{2\beta}{\epsilon_S}\log(1/\delta)\right)$. Then w.p. at least $1 - \delta$,

$$\|\boldsymbol{\theta}_0 - \boldsymbol{\theta}^*\| \leq \frac{1}{\lambda}(\mathcal{O}(\log 1/\alpha) + \epsilon_S).$$

*Proof.* Define $\overline{\ell}^{\mathrm{untr}}(\boldsymbol{\theta}) := \mathbb{E}_{\mathbf{x} \sim p_{\boldsymbol{\theta}_0}}[-\log p_{\boldsymbol{\theta}}(\mathbf{x})]$. Its gradient and Hessian calculations can be done similarly to $\overline{\ell}(\boldsymbol{\theta})$, the truncated version, but with $S = \mathcal{X}$ the full support of the distribution.

Since $E_{\mathbf{z} \sim p_{\boldsymbol{\theta}^*}}[T(\mathbf{z})] - \mathbb{E}_{\mathbf{x} \sim p_{\boldsymbol{\theta}_0}}[T(\mathbf{x})] = \nabla \overline{\ell}(\boldsymbol{\theta}^*)^{\mathrm{untr}}$ is the gradient of the untruncated negative log-likelihood whose optimum is at $\boldsymbol{\theta}_0$, by A1 this gives

$$\|\nabla\overline{\ell}^{\mathrm{untr}}(\boldsymbol{\theta}^*) - \underbrace{\nabla\overline{\ell}^{\mathrm{untr}}(\boldsymbol{\theta}_0)}_{0}\| \geq \lambda\|\boldsymbol{\theta}_0 - \boldsymbol{\theta}^*\| \Rightarrow \|\boldsymbol{\theta}_0 - \boldsymbol{\theta}^*\| \leq \frac{1}{\lambda}\|\nabla\overline{\ell}^{\mathrm{untr}}(\boldsymbol{\theta}^*)\|$$

where the result follows from the fact that $\mathbb{E}_{\mathbf{z} \sim p_{\boldsymbol{\theta}_0}}[T(\mathbf{z})] = \overline{T}$ and $\|\mathbb{E}_{\mathbf{z} \sim p_{\boldsymbol{\theta}^*}}[T(\mathbf{z})] - \overline{T}\| \leq \mathcal{O}(\log 1/\alpha + \epsilon_S)$ w.p. $1 - \delta$ from Lemma 3.7. $\qquad\square$

Note that this result combined with Lemma 3.4 gives Corollary 3.9.

### 3.4 Analysis of Projected Stochastic Gradient Descent Algorithm

Now we have all the tools we need to analyze the main algorithm. For ease of notation, define $d(\alpha) := \epsilon_S + \mathcal{O}(\log 1/\alpha)$ which is a constant that depends on $\alpha$. The following describes the projected stochastic gradient descent algorithm referenced by Theorem 3.1.

---

**Algorithm 1** Projected SGD Algorithm Given Truncated Samples

---

Given $\{\mathbf{x}_i\}_{i=1}^n$, each $\mathbf{x}_i \sim p_{\boldsymbol{\theta}^*}^S$
Initial $\boldsymbol{\theta}_0 \in \mathbb{R}^k$ s.t. $\mathbb{E}_{\mathbf{z} \sim p_{\boldsymbol{\theta}_0}}[T(\mathbf{z})] = \overline{T}$, where $\overline{T} = \frac{1}{n}\sum_i T(\mathbf{x}_i)$.
**for** $i = 0, \ldots, N$ **do**
    $\mathbf{v}_i = \text{SampleGradient}(\mathbf{x}_i, \boldsymbol{\theta}_i)$
    $\boldsymbol{\theta}_{i+1} \leftarrow \boldsymbol{\theta}_i - \eta\mathbf{v}_i$
    Project $\boldsymbol{\theta}_{i+1}$ onto $K = B(\boldsymbol{\theta}_0, \frac{d(\alpha)}{\lambda}) \cap \Theta$.
**end for**
Return $\boldsymbol{\theta}_T$

---

**Algorithm 2** SampleGradient

---

Input: $\mathbf{x}, \boldsymbol{\theta}$
**while** True **do**
    Sample $\mathbf{z} \sim p_{\boldsymbol{\theta}}$
    **if** $\mathbb{1}\{\mathbf{z} \in S\}$ via membership oracle **then**
        Return $T(\mathbf{z}) - T(\mathbf{x})$
    **end if**
**end while**

---

Given the results from the previous sections, we can now prove the main result. The analysis is based on that of Chapter 5 (Theorem 5.7) of [20] which we modify and state below:

**Theorem 3.10** (SGD Convergence). Let $f$ be a $\lambda$-strongly convex function. Let $\boldsymbol{\theta}^* \in \arg\min_{\boldsymbol{\theta} \in K} f(\boldsymbol{\theta})$. Consider the sequence $\{\boldsymbol{\theta}_t\}_{t=1}^N$ generated by SGD (Algorithm 3) and $\{\mathbf{v}_t\}_{t=1}^N$ the sequence random vectors satisfying $\mathbb{E}[\mathbf{v}_t \mid \boldsymbol{\theta}_t] = \nabla f(\boldsymbol{\theta}_t)$ and $\mathbb{E}[\|\mathbf{v}_t\|^2 \mid \boldsymbol{\theta}_t] < \rho^2$ for all $t$, with a constant step size $\eta$ satisfying $0 < \eta < \frac{1}{\lambda}$. It follows that for $t \geq 0$,

$$\mathbb{E}\|\boldsymbol{\theta}_t - \boldsymbol{\theta}^*\|^2 \leq (1 - 2\eta\lambda)^t\|\boldsymbol{\theta}_0 - \boldsymbol{\theta}^*\|^2 + \frac{\eta}{\lambda}\rho^2.$$

The proof is adapted and reproduced in Appendix D.1 for completeness. To apply the above theorem we need to take care of statistical problems:

(i) **strong convexity** $f$ is a strongly convex function over $K$ a convex set

(ii) **smoothness** $f$ is a Lipschitz-smooth function over $K$

(iii) **feasibility of optimal solution** $\boldsymbol{\theta}^* \in K$.

(iv) **bounded variance step** for all $t$, $\mathbb{E}[\|\mathbf{v}_t\|^2 \mid \boldsymbol{\theta}_t] < \rho^2$ for some $\rho^2$

and algorithmic ones:

(a) **initial feasible point** efficiently compute an initial feasible point $\boldsymbol{\theta}_0$

(b) **unbiased gradient estimation** efficiently sample an unbiased estimate of $\nabla f$ $(=\nabla\bar{\ell})$

(c) **efficient projection** efficiently project to parameter space $K$

**Statistical problems.** Firstly, (iii) is assumed. (i-ii) is addressed by Lemmas 3.2, 3.3, 3.6, 3.9, 3.7, 3.4. In particular, we can initialize with $\boldsymbol{\theta}_0$ such that $\|\boldsymbol{\theta}_0 - \boldsymbol{\theta}^*\| \leq \frac{d(\alpha)}{\lambda}$ by Lemmas 3.6, 3.9, 3.7, with probability at least $1 - \delta$. Given this $\boldsymbol{\theta}_0$, we can construct $K = B(\boldsymbol{\theta}_0, \frac{d(\alpha)}{\lambda}) \cap \Theta$ which has the property that

$$\|\boldsymbol{\theta} - \boldsymbol{\theta}^*\| \leq \frac{2}{\lambda}d(\alpha), \ \ \forall \boldsymbol{\theta} \in K.$$

Thus by Lemma 3.4, we will also have

$$p_{\boldsymbol{\theta}}(S) \geq \alpha^2 \exp\left(-6\frac{\kappa}{\lambda} \cdot (d(\alpha))^2\right) > 0, \ \ \forall \boldsymbol{\theta} \in K$$

where $\kappa = L/\lambda$ is the condition number. Since we are projecting to $K$ in which all parameters have non-trivial mass, our objective remains strongly convex. In particular, our objective $\bar{\ell}(\boldsymbol{\theta})$ has

$$\lambda_S I \preceq \nabla^2\bar{\ell}(\boldsymbol{\theta}) \preceq L_S I, \ \ \forall \boldsymbol{\theta} \in K,$$

where $\lambda_S = \frac{1}{2}\left(\frac{\alpha^2 \exp\left(-6\frac{\kappa}{\lambda}\cdot(d(\alpha))^2\right)}{4C\deg}\right)^{2\deg} \lambda$ and $L_S = \frac{\exp\left(6\frac{\kappa}{\lambda}\cdot(d(\alpha))^2\right)}{\alpha^2} L$ are some constants which depend on $\alpha$, $\lambda$, $L$, and the maximum degree, deg, of the sufficient statistics. It remains to address (iv), which is done by the following lemma.

**Lemma 3.11** (Bounded variance step). Let $\mathbf{v}_i$ denote the output of SampleGradient($\mathbf{x}_i, \boldsymbol{\theta}_i$) at any iteration $i \in [N]$. Provided that $\|\mathbb{E}^S_{p_{\boldsymbol{\theta}}}[T(\mathbf{x})] - \mathbb{E}_{p_{\boldsymbol{\theta}}}[T(\mathbf{x})]\| \leq \mathcal{O}(\log 1/p_{\boldsymbol{\theta}}(S))$ for all $\boldsymbol{\theta} \in K$,

$$\mathbb{E}[\|\mathbf{v}_i \mid \boldsymbol{\theta}_i\|^2] \leq kL_S + kL + (1 + 2\kappa)^2(\mathcal{O}(\log 1/p_{\boldsymbol{\theta}_i}(S)))^2.$$

*Proof.* At any iteration $i$ (arbitrary),

$$\mathbb{E}[\|\mathbf{v}_i\|^2 \mid \boldsymbol{\theta}_i] = \mathbb{E}_{(\mathbf{z},\mathbf{x})\sim p^S_{\boldsymbol{\theta}_i}\otimes p^S_{\boldsymbol{\theta}*}}\left[\|T(\mathbf{z}) - T(\mathbf{x})\|^2\right]$$

$$= \mathbb{E}_{(\mathbf{z},\mathbf{x})\sim p^S_{\boldsymbol{\theta}_i}\otimes p^S_{\boldsymbol{\theta}*}}\left[\|T(\mathbf{z})\|^2 - 2\langle T(\mathbf{z}), T(\mathbf{x})\rangle + \|T(\mathbf{x})\|^2\right]$$

$$= \mathrm{Tr}\left(\mathbf{Cov}[T(\mathbf{z})]\right) + (\mathbb{E}[\|T(\mathbf{z})\|])^2 + \mathrm{Tr}\left(\mathbf{Cov}[T(\mathbf{x})]\right) + (\mathbb{E}[\|T(\mathbf{x})\|])^2 - 2\langle\mathbb{E}[T(\mathbf{z})], \mathbb{E}[T(\mathbf{x})]\rangle$$

$$= \mathrm{Tr}\left(\mathbf{Cov}[T(\mathbf{z})]\right) + \mathrm{Tr}\left(\mathbf{Cov}[T(\mathbf{x})]\right) + \|\mathbb{E}_{p^S_{\boldsymbol{\theta}_i}}[T(\mathbf{z})] - \mathbb{E}_{p^S_{\boldsymbol{\theta}*}}[T(\mathbf{x})]\|^2$$

$$\leq kL_S + kL + (1 + 2\kappa)^2(\mathcal{O}(\log 1/p_{\boldsymbol{\theta}_i}(S)))^2$$

In the last step, we've used the fact that

$$\|\mathbb{E}_{p^S_{\boldsymbol{\theta}_i}}[T(\mathbf{z})] - \mathbb{E}_{p^S_{\boldsymbol{\theta}*}}[T(\mathbf{x})]\| \leq \|\mathbb{E}_{p^S_{\boldsymbol{\theta}_i}}[T(\mathbf{z})] - \mathbb{E}_{p_{\boldsymbol{\theta}_i}}[T(\mathbf{z})]\| + \|\mathbb{E}_{p_{\boldsymbol{\theta}_i}}[T(\mathbf{z})] - \mathbb{E}_{p^S_{\boldsymbol{\theta}*}}[T(\mathbf{z})]\|$$

$$= \|\mathbb{E}_{p^S_{\boldsymbol{\theta}_i}}[T(\mathbf{z})] - \mathbb{E}_{p_{\boldsymbol{\theta}_i}}[T(\mathbf{z})]\| + \|\mathbb{E}_{p_{\boldsymbol{\theta}_i}}[T(\mathbf{z})] - \mathbb{E}_{p_{\boldsymbol{\theta}_0}}[T(\mathbf{z})]\|$$

$$\leq \mathcal{O}(\log 1/p_{\boldsymbol{\theta}_i}(S)) + (2L/\lambda)(\mathcal{O}(\log 1/p_{\boldsymbol{\theta}_i}(S)) + \epsilon_S)$$

by assumption, smoothness, Cor. 3.8 and Lemma 3.9. $\qquad\square$

**Algorithmic problems.** For the algorithmic problems, by Cor. 3.8 and Lemmas 3.9, 3.7, we can address (a) by solving the empirical MLE problem with no truncation. Given that we can efficiently sample exactly (or approximately see Appendix D.2) from the non-truncated $p_{\boldsymbol{\theta}}$ for any $\boldsymbol{\theta}$, we can sample unbiased gradients via Algorithm 2 with expected $\mathcal{O}(1/p_{\boldsymbol{\theta}_i}(S)) = \mathcal{O}\left(\frac{\exp(6\kappa \cdot (d(\alpha))^2)}{\alpha^2}\right)$ samples at each step $t$ to address (b). Point (c) can be done efficiently, since our parameter space is a simple intersection of Euclidean balls if we choose $\Theta$ to be a Euclidean ball that sits inside the whole parameter space which contains $\boldsymbol{\theta}^*$.

Let $D(k, L, \lambda, \alpha) = \frac{k(L_S+L)+(1+2\kappa)^2(6\kappa(d(\alpha))^2 - 2\log\alpha)^2}{\lambda_S^2\epsilon^2}$. Putting everything together, to get $\mathbb{E}\|\boldsymbol{\theta}_N - \boldsymbol{\theta}^*\|^2 \leq \epsilon^2$, the number of iterations and samples should be

$$N \geq \max\left\{D(k, L, \lambda, \alpha), \frac{1}{2}\right\} \log\left(\frac{2d(\alpha)}{\lambda\epsilon^2}\right),$$

provided that $\eta = \min\{\frac{\lambda_S\epsilon^2}{2\rho^2}, \frac{1}{\lambda_S}\}$, applying Lemma D.1 to the bound from Theorem 3.10 with $A = \frac{\rho^2}{\lambda_S}$, $C = \lambda_S$, $\mu = 2\lambda_S$, and $\rho^2 = kL_S + kL + (1+2\kappa)^2(\mathcal{O}(\log 1/p_{\boldsymbol{\theta}}(S)))^2$ by Lemma 3.11. Further, Lemma 3.4 guarantees $\mathcal{O}(\log 1/p_{\boldsymbol{\theta}}(S)) \leq \mathcal{O}\left(\log\frac{\exp(6\kappa\cdot(d(\alpha))^2)}{\alpha^2}\right)$ for all $\boldsymbol{\theta} \in K$.

In probability, we get $\mathbb{P}(\|\boldsymbol{\theta}_N - \boldsymbol{\theta}^*\|^2 \geq 3\epsilon^2) \leq 1/3$ by Markov's inequality. Then we can amplify the probability of success to $1 - \delta$ by repeating the procedure from scratch $\log 1/\delta$ times, as in [10]. Given a polynomial time ($\text{poly}(m, k, 1/\epsilon)$) algorithm $A_S$ to sample from $p_{\boldsymbol{\theta}}$ for all $\boldsymbol{\theta}$, each iteration takes expected $\mathcal{O}(1/p_{\boldsymbol{\theta}_i}(S)) = \mathcal{O}\left(\frac{\exp(6\frac{\kappa}{\lambda}\cdot(d(\alpha))^2)}{\alpha^2}\right)$ times the running time of $A_S$ plus the projection step (which is also efficient). This completes the result of Theorem 3.1.

*Remark.* In the Gaussian case, the sample complexity was given in terms of $m$, the dimension of the $\mathbf{x}$ and was stated in [10] as $\widetilde{\mathcal{O}}(m^2/\epsilon^2)$. For multivariate Gaussian, the dimension $k$ of $\boldsymbol{\theta}$ is the dimension of $[\mathbf{x}, \mathbf{x}\mathbf{x}^\top]$ (vectorized) which is $m + m^2$, thus we can recover the previous result.

**Numerical example.** While a lot of the analyses shown can seem complicated, they give some guarantees for a rather simple algorithm: simply initialize your parameters using MLE as if there is no truncation at all, then run projected stochastic gradient descent using rejection sampling. We do not even need to describe the truncation set $S$ as long as we can query whether a point is inside $S$ or not efficiently. As a proof of concept for the use of this algorithm, we've implemented our simple projected SGD algorithm to learn the parameters of multivariate exponential distributions, given truncated samples. To save space in the main body for exposition of the theoretical results, we've included the results and details in Appendix E.

## 4 Discussion

To our knowledge, this work is the first which develops a computationally and statistically efficient algorithm for learning from samples truncated to very general sets $S$ of the form in [10] in high dimensions whose distribution does not rely on Gaussianity. This work also has interesting implications for learning general log-concave distributions through applying the Taylor theorem to the log density as in [12] and for sampling (e.g., as in [30]). Through generalizing the previous Gaussian results to general exponential families, we also extract the broader abstract properties (e.g., concentration and anti-concentration) of distributions for which the previously proposed projected stochastic gradient descent procedure still applies. It would be interesting to understand how much these results can be generalized to densities beyond exponential families, or even to extend the current work presented from expected polynomial running time to deterministic polynomial running time. We hope that our work provides the foundation for future work in this direction.

## 5 Acknowledgements

We thank Tim Kunisky for insightful discussions during the preparation of this work. Jane Lee was supported by a GFSD fellowship sponsored by the U.S. National Security Agency (NSA).

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

# A Proofs and Calculations Regarding the Objective

## A.1 The Truncated Negative Expected Log-Likelihood Function

The negative log-likelihood that $\mathbf{x} \in S$ is a sample of $p_{\boldsymbol{\theta}}^S(\mathbf{x})$ is

$$\underbrace{\ell(\theta, \mathbf{x})}_{-\log p_{\boldsymbol{\theta}}^S(\mathbf{x})} := -\log h(\mathbf{x}) - \boldsymbol{\theta}^\top T(\mathbf{x}) + \log \int_S h(\mathbf{x}') \exp(\theta^\top T(\mathbf{x}')) d\mathbf{x}'.$$

Its gradient w.r.t. $\boldsymbol{\theta}$ is

$$\nabla \ell(\boldsymbol{\theta}, \mathbf{x}) = -T(\mathbf{x}) + \frac{\int_S T(\mathbf{x}') h(\mathbf{x}') \exp(\theta^\top T(\mathbf{x}')) d\mathbf{x}'}{\int_S h(\mathbf{x}') \exp(\theta^\top T(\mathbf{x}')) d\mathbf{x}'}$$

$$= -T(\mathbf{x}) + \frac{\int_S T(\mathbf{x}') h(\mathbf{x}') \exp(\theta^\top T(\mathbf{x}') - A(\boldsymbol{\theta})) d\mathbf{x}'}{\int_S h(\mathbf{x}') \exp(\theta^\top T(\mathbf{x}') - A(\boldsymbol{\theta})) d\mathbf{x}'}$$

$$= -T(\mathbf{x}) + \mathbb{E}_{\mathbf{z} \sim p_{\boldsymbol{\theta}}^S}[T(\mathbf{z})]$$

The Hessian is

$$\nabla^2 \ell(\boldsymbol{\theta}) = \frac{(\int_S T(\mathbf{x}) T(\mathbf{x})^\top h(\mathbf{x}) \exp(\boldsymbol{\theta}^\top T(\mathbf{x}) - A(\boldsymbol{\theta})) d\mathbf{x})}{(\int_S h(\mathbf{x}) \exp(\boldsymbol{\theta}^\top T(\mathbf{x}) - A(\boldsymbol{\theta})) d\mathbf{x})}$$

$$- \frac{(\int_S T(\mathbf{x}) h(\mathbf{x}) \exp(\boldsymbol{\theta}^\top T(\mathbf{x}) - A(\boldsymbol{\theta})) d\mathbf{x})}{(\int_S h(\mathbf{x}) \exp(\boldsymbol{\theta}^\top T(\mathbf{x}) - A(\boldsymbol{\theta})) d\mathbf{x})} \cdot \left( \frac{(\int_S T(\mathbf{x}) h(\mathbf{x}) \exp(\boldsymbol{\theta}^\top T(\mathbf{x}) - A(\boldsymbol{\theta})) d\mathbf{x})}{(\int_S h(\mathbf{x}) \exp(\boldsymbol{\theta}^\top T(\mathbf{x}) - A(\boldsymbol{\theta})) d\mathbf{x})} \right)^\top$$

$$= \mathbf{Cov}_{\mathbf{x} \sim p_{\boldsymbol{\theta}}^S}[T(\mathbf{x}), T(\mathbf{x})]$$

We can similarly define the population negative log-likelihood as

$$\bar{\ell}(\boldsymbol{\theta}) := \mathbb{E}_{\mathbf{x} \sim p_{\boldsymbol{\theta}^*}^S} \left[ -\log h(\mathbf{x}) - \boldsymbol{\theta}^\top T(\mathbf{x}) \right] + \log \int_S h(\mathbf{x}) \exp(\theta^\top T(\mathbf{x})) d\mathbf{x}),$$

$$\nabla \bar{\ell}(\boldsymbol{\theta}) = \mathbb{E}_{\mathbf{x} \sim p_{\boldsymbol{\theta}^*}^S}[-T(\mathbf{x})] + \mathbb{E}_{\mathbf{x} \sim p_{\boldsymbol{\theta}}^S}[T(\mathbf{x})],$$

$$\nabla^2 \bar{\ell}(\boldsymbol{\theta}) = \nabla^2 \ell(\boldsymbol{\theta})$$

## A.2 Proof of Lemma 3.2

*Proof.* Define the following quantities:

$$\mathbf{R}^* = \mathbb{E}_{\mathbf{x} \sim p_{\boldsymbol{\theta}}} \left[ (T(\mathbf{x}) - \mathbb{E}_{\mathbf{x} \sim p_{\boldsymbol{\theta}}}[T(\mathbf{x})]) \cdot (T(\mathbf{x}) - \mathbb{E}_{\mathbf{x} \sim p_{\boldsymbol{\theta}}}[T(x)])^\top \right]$$

$$\mathbf{R}' = \mathbb{E}_{x \sim p_{\boldsymbol{\theta}}} \left[ \left( T(\mathbf{x}) - \mathbb{E}_{\mathbf{x} \sim p_{\boldsymbol{\theta}}^S}[T(\mathbf{x})] \right) \cdot \left( T(\mathbf{x}) - \mathbb{E}_{\mathbf{x} \sim p_{\boldsymbol{\theta}}^S}[T(\mathbf{x})] \right)^\top \right]$$

$$\mathbf{R} = \mathbb{E}_{\mathbf{x} \sim p_{\boldsymbol{\theta}}^S} \left[ \left( T(\mathbf{x}) - \mathbb{E}_{\mathbf{x} \sim p_{\boldsymbol{\theta}}^S}[T(\mathbf{x})] \right) \cdot \left( T(\mathbf{x}) - \mathbb{E}_{\mathbf{x} \sim p_{\boldsymbol{\theta}}^S}[T(\mathbf{x})] \right)^\top \right]$$

**Claim 2.** $R' \succeq R^*$. (Proof in Appendix A.4.)

Now, let $\xi \in \mathbb{R}^k$ with $\|\xi\|_2^2 = 1$ arbitrary. Then

$$\xi^\top \mathbf{R}^* \xi = \xi^\top \mathbb{E}_{\mathbf{x} \sim p_{\boldsymbol{\theta}}} \left[ (T(\mathbf{x}) - \mathbb{E}_{\mathbf{x} \sim p_{\boldsymbol{\theta}}}[T(\mathbf{x})]) \cdot (T(\mathbf{x}) - \mathbb{E}_{\mathbf{x} \sim p_{\boldsymbol{\theta}}}[T(\mathbf{x})])^\top \right] \xi = \mathbb{E}_{\mathbf{x} \sim p_{\boldsymbol{\theta}}} \left[ p_\xi(\mathbf{x}) \right]$$

$$\xi^\top \mathbf{R}' \xi = \mathbb{E}_{\mathbf{x} \sim p_{\boldsymbol{\theta}}} \left[ p_\xi'(\mathbf{x}) \right]$$

$$\xi^\top \mathbf{R} \xi = \mathbb{E}_{\mathbf{x} \sim p_{\boldsymbol{\theta}}^S} \left[ p_\xi'(\mathbf{x}) \right]$$

where $p_\xi(\mathbf{x}), p_\xi'(\mathbf{x})$ are polynomials of degree at most $2d$ whose coefficients depend on $\xi$ (under A3). Furthermore, note that for any $\xi \in \mathbb{R}^k$, $p_\xi(\mathbf{x}) \geq 0$ and $p_\xi'(\mathbf{x}) \geq 0$ (due to the rank one matrix inside the expectation being PSD).

First, since $\mathbf{R}' \succeq \mathbf{R}^* \iff \xi^\top \mathbf{R}' \xi \geq \xi^\top \mathbf{R}^* \xi$, we have

$$\mathbb{E}_{\mathbf{z} \sim p_\theta} \left[ p'_\xi(\mathbf{z}) \right] \geq \mathbb{E}_{\mathbf{z} \sim p_\theta} \left[ p_\xi(\mathbf{z}) \right] \geq \lambda.$$

Now define the set $A := \{\mathbf{x} : p'_\xi(\mathbf{x}) \leq \gamma\}$ for $\gamma = \left( \frac{\beta}{4Cd} \right)^{2d} \lambda$ where $p_\theta(S) = \beta > 0$. Theorem 8 of [5] says

$$p_\theta(A) \leq \frac{Cq\gamma^{1/(2d)}}{\left( \mathbb{E}_{\mathbf{z} \sim p_\theta} \left[ p'_\xi(\mathbf{z})^{q/2d} \right]^{1/q} \right)} \overset{q=2d}{=} \frac{2Cd\gamma^{1/(2d)}}{\left( \underbrace{\mathbb{E}_{\mathbf{z} \sim p_\theta} \left[ p'_\xi(\mathbf{z}) \right]}_{\geq \lambda} \right)^{1/(2d)}} \leq \frac{2Cd \cdot \gamma^{1/(2d)}}{\lambda^{1/(2d)}} = \frac{\beta}{2}.$$

Now we can split $\mathbb{E}_{\mathbf{z} \sim p_\theta^S} \left[ p'_\xi(\mathbf{z}) \right]$ into the part on $S \cap A$ and $S \cap A^c$. Note that if $p_\theta(S) = \beta$ and $p_\theta(A) \leq \frac{\beta}{2}$, this implies $p_\theta(S \cap A^c) \geq \frac{\beta}{2}$ as

$$p_\theta(S \cap A^c) \geq p_\theta(S) + p_\theta(A^c) - p_\theta(S \cup A^c) \geq \beta + \left( 1 - \frac{\beta}{2} \right) - 1 = \frac{\beta}{2}.$$

Then

$$\mathbb{E}_{\mathbf{z} \sim p_\theta^{S \cap A}} \left[ p'_\xi(\mathbf{z}) \right] + \mathbb{E}_{\mathbf{z} \sim p_\theta^{S \cap A^c}} \left[ p'_\xi(\mathbf{z}) \right] \geq \frac{p_\theta(S \cap A)}{p_\theta(S)} \cdot 0 + \frac{p_\theta(S \cap A^c)}{p_\theta(S)} \cdot \gamma \geq \frac{1}{2}\gamma \Rightarrow \mathbb{E}_{\mathbf{z} \sim p_\theta^S} \left[ p'_\xi(\mathbf{z}) \right] \geq \frac{1}{2} \left( \frac{\beta}{4Cd} \right)^{2d} \lambda$$

and the claim follows. $\square$

## A.3 Proof of Lemma 3.3

*Proof.* Similar to the proof of the previous lemma, define the following quantities:

$$\mathbf{R}^* = \mathbb{E}_{\mathbf{x} \sim p_\theta} \left[ (T(\mathbf{x}) - \mathbb{E}_{\mathbf{x} \sim p_\theta}[T(\mathbf{x})]) \cdot (T(\mathbf{x}) - \mathbb{E}_{\mathbf{x} \sim p_\theta}[T(\mathbf{x})])^\top \right]$$

$$\mathbf{R}'' = \mathbb{E}_{\mathbf{x} \sim p_\theta^S} \left[ (T(\mathbf{x}) - \mathbb{E}_{\mathbf{x} \sim p_\theta}[T(\mathbf{x})]) \cdot (T(\mathbf{x}) - \mathbb{E}_{\mathbf{x} \sim p_\theta}[T(\mathbf{x})])^\top \right]$$

$$\mathbf{R} = \mathbb{E}_{\mathbf{x} \sim p_\theta^S} \left[ \left( T(\mathbf{x}) - \mathbb{E}_{\mathbf{x} \sim p_\theta^S}[T(\mathbf{x})] \right) \cdot \left( T(\mathbf{x}) - \mathbb{E}_{\mathbf{x} \sim p_\theta^S}[T(\mathbf{x})] \right)^\top \right]$$

**Claim 3.** It holds that $\mathbf{R}'' \succeq \mathbf{R}$. (Similar proof to Claim 2.)

Let $\xi \in \mathbb{R}^k$ with $\|\xi\|_2^2 = 1$ arbitrary. Then

$$\xi^\top \mathbf{R}^* \xi = \mathbb{E}_{\mathbf{x} \sim p_\theta}[f_\xi(\mathbf{x})]$$
$$\xi^\top \mathbf{R}'' \xi = \mathbb{E}_{\mathbf{x} \sim p_\theta^S}[f_\xi(\mathbf{x})]$$
$$\xi^\top \mathbf{R} \xi = \mathbb{E}_{\mathbf{x} \sim p_\theta^S}[f'_\xi(\mathbf{x})]$$

where $f_\xi(\mathbf{x}), f'_\xi(\mathbf{x})$ are some functions which depend on $\mathbf{x}$ and $\xi$ (e.g., polynomials of degree at most $2d$ under A3). By the previous claim, we also have

$$\mathbb{E}_{\mathbf{x} \sim p_\theta^S}[f_\xi(\mathbf{x})] \geq \mathbb{E}_{\mathbf{x} \sim p_\theta^S}[f'_\xi(\mathbf{x})].$$

Note that

$$\mathbb{E}_{\mathbf{x} \sim p_\theta^S}[f_\xi(\mathbf{x})] = \int_\mathcal{X} p_\theta^S(\mathbf{x}) \cdot f_\xi(\mathbf{x}) d\mathbf{x} = \int_\mathcal{X} \frac{1}{p_\theta(S)} p_\theta(\mathbf{x}) \cdot f_\xi(\mathbf{x}) \cdot \mathbb{1}\{\mathbf{x} \in S\} d\mathbf{x} \leq \frac{1}{p_\theta(S)} \underbrace{\int p_\theta(\mathbf{x}) f_\xi(\mathbf{x}) d\mathbf{x}}_{=\mathbb{E}_{\mathbf{x} \sim p_\theta}[f_\xi(\mathbf{x})]}.$$

Since $\lambda I \preceq \mathbf{R}^* \preceq LI$ by A1, it holds that $\xi^\top \mathbf{R}^* \xi = \mathbb{E}_{\mathbf{x} \sim p_\theta}[f_\xi(\mathbf{x})] \leq L$, thus the following inequalities hold:

$$\xi^\top \mathbf{R} \xi = \mathbb{E}_{\mathbf{x} \sim p_\theta^S}[f'_\xi(\mathbf{x})] \leq \mathbb{E}_{\mathbf{x} \sim p_\theta^S}[f_\xi(\mathbf{x})] \leq \frac{1}{p_\theta(S)} L.$$

$\square$

## A.4 Proof of Claim 2

We will prove a general claim which should take care of both claims in Lemmas 3.2 and 3.3.

**Claim 4.** Let $\mathbf{x} \sim \rho$ be a random vector with mean $\boldsymbol{\mu}$. Let $\mathbf{b}$ be another vector such that $\mathbf{b} \neq \boldsymbol{\mu}$. Then

$$\mathbf{Cov}_{\mathbf{x} \sim \rho}[\mathbf{x}, \mathbf{x}] = \mathbb{E}_{\mathbf{x} \sim \rho}[(\mathbf{x} - \boldsymbol{\mu})(\mathbf{x} - \boldsymbol{\mu})^\top] = \mathbb{E}_{\mathbf{x} \sim \rho}[(\mathbf{x} - \mathbf{b})(\mathbf{x} - \mathbf{b})^\top] - (\mathbf{b} - \boldsymbol{\mu})(\mathbf{b} - \boldsymbol{\mu})^\top.$$

*Proof.*

$$\mathbb{E}[(\mathbf{x} - \boldsymbol{\mu})(\mathbf{x} - \boldsymbol{\mu})^\top]$$
$$= \mathbb{E}[(\mathbf{x} - \mathbf{b} + \mathbf{b} - \boldsymbol{\mu})(\mathbf{x} - \mathbf{b} + \mathbf{b} - \boldsymbol{\mu})^\top]$$
$$= \mathbb{E}[(\mathbf{x} - \mathbf{b})(\mathbf{x} - \mathbf{b})^\top] + \underbrace{\mathbb{E}[(\mathbf{x} - \mathbf{b})(\mathbf{b} - \boldsymbol{\mu})^\top]}_{=(-1) \cdot \mathbb{E}[(\mathbf{b} - \boldsymbol{\mu})(\mathbf{b} - \boldsymbol{\mu})^\top]} + \underbrace{\mathbb{E}[(\mathbf{b} - \boldsymbol{\mu})(\mathbf{x} - \mathbf{b})^\top]}_{=(-1) \cdot \mathbb{E}[(\mathbf{b} - \boldsymbol{\mu})(\mathbf{b} - \boldsymbol{\mu})^\top]} + \underbrace{\mathbb{E}[(\mathbf{b} - \boldsymbol{\mu})(\mathbf{b} - \boldsymbol{\mu})^\top]}_{=\mathbb{E}[(\mathbf{b} - \boldsymbol{\mu})(\mathbf{b} - \boldsymbol{\mu})^\top]}$$
$$= \mathbb{E}[(\mathbf{x} - \mathbf{b})(\mathbf{x} - \mathbf{b})^\top] - \mathbb{E}[(\mathbf{b} - \boldsymbol{\mu})(\mathbf{b} - \boldsymbol{\mu})^\top]$$

$\square$

As a corollary, since the second term is a rank-1 matrix (thus PSD), we have that $\mathbb{E}[(\mathbf{x} - \mathbf{b})(\mathbf{x} - \mathbf{b})^\top] \succeq \mathbb{E}[(\mathbf{x} - \boldsymbol{\mu})(\mathbf{x} - \boldsymbol{\mu})^\top]$.

## B Examples of Other Distributions which Satisfy Assumptions

**Example 1** (Exponential Distribution). The exponential distribution density can be written

$$p_\lambda(x) = \lambda \exp(-\lambda x) = \exp(-\lambda x + \log(\lambda)),$$

defined on $x \in \mathbb{R}^+$ which is a convex set and for $\lambda > 0$. In natural form, it is

$$p_\theta(x) = \exp(\theta x + \log(-\theta))),$$

defined for $\theta < 0$. Note that

- $T(x) = x$ is a polynomial in $x$.

- This is log-linear in $x$ (so log-concave in $x$).

- Variance of the sufficient statistic is simply the variance, which is $1/\theta^2 > 0$ for any $\theta < 0$. If we restrict $\theta$ in a bounded set, the negative log-likelihood will be strongly convex and smooth in $\theta$.

**Example 2** (Weibull Distribution with known shape $k$). The Weibull distribution with known shape $k > 0$ has density

$$p_\lambda(x) = \exp\left((k-1)\log x + \left(-\frac{1}{\lambda^k}\right)x^k + \log k - k \log \lambda\right)$$

defined on $x \in \mathbb{R}^+$ and $\lambda > 0$. We can re-parameterize this in terms of $\theta = -\frac{1}{\lambda^k}$ with $\theta < 0$ as

$$p_\theta(x) = x^{k-1} \exp(\theta \cdot x^k + \log k + \log(-\theta)).$$

Then

- $T(x) = x^k$ is polynomial in $x$.

- $p_\theta(x)$ is log-concave in $x$ if $k > 1$ (where recall $x \in \mathbb{R}^+$ and $\theta < 0$).

- The variance of the sufficient statistic can also be found by taking the second derivative of $A(\theta) = -\log k - \log(-\theta)$ w.r.t. $\theta$, which is also $1/\theta^2 > 0$.

**Example 3** (Continuous Bernoulli). The continuous Bernoulli density [31] can be written

$$p_\lambda(x) = \exp\left( \log \frac{\lambda}{1 - \lambda} - \log \frac{1 - 2\lambda}{(1 - \lambda) \log \frac{1-\lambda}{\lambda}} \right)$$

with support $x \in [0, 1]$ and $\lambda \in (0, 1)$. We can re-parameterize this in terms of $\theta = \log \frac{\lambda}{1-\lambda}$ with $\theta \in [0, \infty)$ so

$$p_\theta(x) = \exp\left( \theta x - \log \frac{e^\theta - 1}{\theta} \right).$$

Then

- $T(x) = x$ is polynomial in $x$.

- $p_\theta(x)$ is log-linear in $x$ (so log-concave).

- The variance of sufficient statistic is simply the variance again, which is given by

$$\mathbf{Var}(X) = \begin{cases} 1/12 & \text{if } \lambda = 1/2 \\ \frac{(\lambda-1)\lambda}{(1-2\lambda)^2} + \frac{1}{(2\tanh^{-1}(1-2\lambda))^2} & \text{otherwise} \end{cases}$$

  This is strictly positive and bounded for all values of $\lambda$ (thus all values of $\theta$).

**Example 4** (Continuous Poisson). A continuous version of the Poisson distribution (although there can be others [23]) can be written

$$p_\lambda(x) = \frac{1}{Z(\lambda)} \frac{e^{-\lambda} \lambda^x}{\Gamma(x + 1)}$$

with support $x \in [0, \infty)$ and $\lambda \in (0, \infty)$. We can write this with $\theta = \log \lambda$ so

$$p_\theta(x) = \frac{1}{\Gamma(x + 1)} \exp(\theta x - A(\theta)).$$

Then

- $T(x) = x$ is polynomial in $x$.

- $p_\theta(x)$ is log-concave in $x$ for $x \in \mathbb{R}^+$.

- In $\lambda$ parameters, the mean of this distribution is $\lambda$ through usual calculations (e.g., similar to those of the Gamma distribution). Note: we can absorb the $e^{-\lambda}$ term into the partition function.

$$\begin{aligned} \mathbb{E}[X] &= \frac{1}{Z(\lambda)} \int_0^\infty \frac{x \lambda^x}{\Gamma(x + 1)} dx \\ &= \frac{1}{Z(\lambda)} \int_0^\infty \frac{x \lambda^x}{x \cdot \Gamma(x)} dx & \Gamma(x + 1) = x \cdot \Gamma(x) \\ &= \frac{\lambda}{Z(\lambda)} \int_1^\infty \frac{\lambda^{x-1}}{\Gamma(x)} dx & \text{Partition function, change var. } z = x - 1 \\ &= \lambda \end{aligned}$$

  Similarly, we should be able to show the variance is $\lambda$ as usual. In $\theta$ space, this means the variance is $\exp(\theta)$ for $\theta \in \mathbb{R}$ which is always positive. Again, we can make it bounded by restricting $\theta$ to some set.

**Example 5** (Multivariate Gaussian). The multivariate Gaussian also satisfies all of these properties. Recall that the sufficient statistics of the multivariate Gaussian has

- $T(\mathbf{x}) = [\mathbf{x}, \mathbf{x}\mathbf{x}^\top]$ is a polynomial in the components of $\mathbf{x}$ with degree at most 2 (where the $\mathbf{x}\mathbf{x}^\top$ term can be thought of as the vector after standard vectorization).

- The multivariate Gaussian density is strongly log-concave.

- The covariance matrix (of the sufficient statistics) has a complicated form, which the authors of [10] have analyzed the lower bound for, e.g., in their Claims 1 and 2. As before, we can restrict our parameter space to ensure upper bounds.

**Example 6** (Generalized Linear Models). This example is the same as the one given in [25] for generalized linear models. It is restated here for completeness.

Consider when we have some covariance, response pair $(X, Y)$ drawn from some distribution $D$. Suppose that we have a family of distributions $P(\cdot \mid \theta; X)$ such that, for each $X$, it is an exponential family with sufficient statistic $t_{y,X}$

$$P(y \mid \theta; X) = h(y) \exp\left(\langle \theta, t_{y,X} \rangle - A(\theta, X)\right).$$

We can consider a one-dimensional exponential family $q_\nu$ with parameterization $\nu = \langle \theta, X \rangle$, then

$$P(y \mid \theta; X) = h(y) \exp\left(y\langle \theta, X \rangle - \log Z(\langle \theta, X \rangle)\right)$$

where we see that $t_{y,X} = yX$ and the log partition function $A(\theta, X) = \log Z(\langle \theta, X \rangle)$. When $q_\nu$ is Bernoulli family or unit variance Gaussian family, this corresponds to *logistic regression* or *least squares regression*, respectively.

We can appropriately generalize this to beyond linear models (e.g., polynomials) provided that we can keep the distribution log-concave.

**Comment on A3.** We mentioned in the main paper that this assumption combined with log-concavity provides the anti-concentration property that we need for Lemma 3.2. We assume it for simplicity of exposition, but it should be noted that as long as we have the type of anti-concentration property to control how much the covariance can shrink under truncation, we do not necessarily need $T(\mathbf{x})$ to be polynomial. However, we've provided examples of exponential families which already satisfy this above (and there are potentially more which can be addressed by this framework that do not have polynomial sufficient statistics but nonetheless exhibit similar anti-concentration properties).

## C  Proofs Relating Truncated and Non-Truncated Quantities

### C.1  General Truncated Densities

Let $\rho$ be a probability distribution on $\mathbb{R}^d$. Let $S \subseteq \mathbb{R}^d$ be such that $\rho(S) = \alpha$ for some $\alpha \in (0, 1]$. Let $\rho^S := \rho(\cdot \mid \cdot \in S)$ be the conditional distribution of $\mathbf{x} \sim \rho$ given that $\mathbf{x} \in S$.

$$\rho^S(\mathbf{x}) = \frac{\rho(\mathbf{x}) \cdot \mathbb{1}\{\mathbf{x} \in S\}}{\rho(S)}.$$

Note that the relative density is

$$\frac{\rho^S(\mathbf{x})}{\rho(\mathbf{x})} = \frac{\mathbb{1}\{\mathbf{x} \in S\}}{\rho(S)}.$$

Then we can compute that the Rényi divergence is a constant for any order $1 \leq q \leq \infty$.

$$\mathsf{KL}(\rho^S \| \rho) = \mathbb{E}_{\rho^S}\left[\log \frac{\rho^S}{\rho}\right] = \mathbb{E}_{\rho^S}\left[\log \frac{1}{\rho(S)}\right] = \log \frac{1}{\alpha}.$$

$$\chi^2(\rho^S \| \rho) = \mathbb{E}_{\rho^S}\left[\frac{\rho^S}{\rho}\right] - 1 = \frac{1}{\rho(S)} - 1 = \frac{1}{\alpha} - 1.$$

$$\mathsf{R}_q(\rho^S \| \rho) = \frac{1}{q-1} \log \mathbb{E}_{\rho^S}\left[\left(\frac{\rho^S}{\rho}\right)^{q-1}\right] = \frac{1}{q-1} \log \frac{1}{\rho(S)^{q-1}} = \log \frac{1}{\rho(S)} = \log \frac{1}{\alpha}.$$

$$\mathsf{R}_\infty(\rho^S \| \rho) = \log \sup_x \frac{\rho^S(x)}{\rho(x)} = \log \frac{1}{\rho(S)} = \log \frac{1}{\alpha}.$$

Note $\mathsf{R}_2(\rho^S \| \rho) = \log(1 + \chi^2(\rho^S \| \rho))$.

We recall the following general estimates.

**Lemma C.1.** For any probability distributions $\rho, \pi$ (such that the quantities below are finite):

1. $\|\mathbb{E}_\rho[\mathbf{x}] - \mathbb{E}_\pi[\mathbf{x}]\| \leq \sqrt{\chi^2(\rho\|\pi)} \cdot \sqrt{\mathbf{Var}_\pi(\mathbf{x})}.$

2. $|\mathbb{E}_\rho[\|\mathbf{x}\|^2] - \mathbb{E}_\pi[\|\mathbf{x}\|^2]| \leq \sqrt{\chi^2(\rho\|\pi)} \cdot \sqrt{\mathbb{E}_\pi[\|\mathbf{x}\|^4]}.$

3. $|\mathbf{Var}_\rho(\mathbf{x}) - \mathbf{Var}_\pi(\mathbf{x})| \leq \sqrt{(\chi^2(\rho\|\pi) + 1)^2 - 1} \cdot \sqrt{2\mathbb{E}_\pi[\|\mathbf{x} - \mathbb{E}_\pi[\mathbf{x}]\|^4]}.$

*Proof.* The first two claims are immediate by Cauchy-Schwarz. For the third one, recall we can write

$$\mathbf{Var}_\rho(\mathbf{x}) = \frac{1}{2}\mathbb{E}_{\rho^{\otimes 2}}[\|\mathbf{x} - \mathbf{y}\|^2].$$

Then by applying part (1) to $\rho^{\otimes 2}$ and $(\pi)^{\otimes 2}$, we get

$$
\begin{aligned}
|\mathbf{Var}_\rho(\mathbf{x}) - \mathbf{Var}_\pi(\mathbf{x})| &\leq \frac{1}{2}\sqrt{\chi^2(\rho^{\otimes 2}\|\pi^{\otimes 2})} \cdot \sqrt{\mathbb{E}_{\pi^{\otimes 2}}[\|\mathbf{x} - \mathbf{y}\|^4]} \\
&= \frac{1}{2}\sqrt{(\chi^2(\rho\|\pi) + 1)^2 - 1} \cdot \sqrt{2\mathbb{E}_\pi[\|\mathbf{x} - \mathbb{E}_\pi[\mathbf{x}]\|^4] + 6\mathbb{E}_\pi[\|\mathbf{x} - \mathbb{E}_\pi[\mathbf{x}]\|^2]^2} \\
&\leq \frac{1}{2}\sqrt{(\chi^2(\rho\|\pi) + 1)^2 - 1} \cdot \sqrt{8\mathbb{E}_\pi[\|\mathbf{x} - \mathbb{E}_\pi[\mathbf{x}]\|^4]}.
\end{aligned}
$$

$\square$

For our application, we have the following. Given a probability distribution $\rho$ on $\mathbb{R}^d$, we let $\mu(\rho) = \mathbb{E}_\rho[\mathbf{x}]$ be its mean, and for $k \in \mathbb{N}$,

$$M_k(\rho) := \mathbb{E}_\rho[\|\mathbf{x} - \mu(\rho)\|^k]^{1/k}.$$

So for example we have $M_2(\rho) = \sqrt{\mathbf{Var}_\rho(\mathbf{x})}$. We also have $M_k(\rho) \leq M_\ell(\rho)$ if $k \leq \ell$.

**Lemma C.2.** Let $\rho$ be a probability distribution on $\mathbb{R}^d$. Let $S \subseteq \mathbb{R}^d$ with $\rho(S) = \alpha \in (0, 1]$. Then

1. $\|\mathbb{E}_{\rho^S}[\mathbf{x}] - \mathbb{E}_\rho[\mathbf{x}]\| \leq \sqrt{\frac{1-\alpha}{\alpha}} \cdot \sqrt{\mathbf{Var}_\rho(\mathbf{x})}.$

2. $|\mathbf{Var}_{\rho^S}(\mathbf{x}) - \mathbf{Var}_\rho(\mathbf{x})| \leq \frac{\sqrt{2(1-\alpha^2)}}{\alpha}M_4(\rho)^2.$

In particular, if $\alpha \in (0, 1]$ is such that $\frac{1}{\alpha^2} \leq 1 + \frac{c^2 M_2(\rho)^4}{2M_4(\rho)^4}$ for some $0 \leq c < 1$, then

$$\mathbf{Var}_{\rho^S}(\mathbf{x}) \geq (1 - c)\mathbf{Var}_\rho(\mathbf{x}).$$

Note that the constraint on $\alpha$ above implies $\frac{1}{\alpha^2} \leq \frac{3}{2}$, so $\alpha \geq \sqrt{2/3}$. But if $M_2(\rho) \ll M_4(\rho)$, then $1 - \alpha$ will be very small.

Recall also that under some conditions, e.g. if $\rho$ is log-concave, then we have the reverse bound that

$$M_2(\rho) \geq C_{2,4}M_4(\rho)$$

for a universal constant $C_{2,4}$, so the constraint above is not too restrictive, as it allows $1 - \alpha$ of constant size.

## C.2 Exponential Families with Strongly Convex and Smooth Log-Partition Functions are Sub-Exponential

Let $\boldsymbol{\theta} \in \Theta$ such that $\boldsymbol{\theta} + \frac{1}{\beta}\mathbf{u} \in \Theta$ for some $\beta > 0$ for all unit vectors $\mathbf{u}$ and such that Assumption A1 holds for $p_{\boldsymbol{\theta}}$. Then $X := \mathbf{u}^\top(T(\mathbf{x}) - \mathbb{E}_{\mathbf{x} \sim p_{\boldsymbol{\theta}}}[T(\mathbf{x})])$ is $SE(L, \beta)$.

*Proof.* WLOG, consider $p_{\boldsymbol{\theta}}$ in the transformed space $\mathbf{x} \mapsto T(\mathbf{x})$ so that

$$p_{\boldsymbol{\theta}}(\mathbf{t}) = h(\mathbf{t})\exp(\boldsymbol{\theta}^\top \mathbf{t} - A(\boldsymbol{\theta}))d\mathbf{t},$$

where $\boldsymbol{\theta} \in \Theta$ and $A(\boldsymbol{\theta}) = \log(Z(\boldsymbol{\theta})) = \log\left(\int_{\mathcal{T}(\mathcal{X})} h(\mathbf{t})\exp(\boldsymbol{\theta}^\top \mathbf{t})d\mathbf{t}\right)$ is the log-partition function. Note that $\nabla^2 A(\boldsymbol{\theta}) = \mathbf{Cov}_{\mathbf{t} \sim p_{\boldsymbol{\theta}}(\mathbf{t})}[\mathbf{t}] = \mathbf{Cov}_{\mathbf{x} \sim p_{\boldsymbol{\theta}}(\mathbf{x})}[T(\mathbf{x})]$, and by A1, $A(\boldsymbol{\theta})$ is a $\lambda$-strongly convex and $L$-smooth function in $\boldsymbol{\theta}$.

To show that $p_{\boldsymbol{\theta}}(\mathbf{t})$ is sub-exponential with parameters $(\nu^2, \beta)$ we need to show that its moment generating function satisfies $\mathbb{E}[e^{\gamma \mathbf{u}^\top (\mathbf{t} - \boldsymbol{\mu})}] \leq e^{\gamma^2 \nu^2/2}$, where $\boldsymbol{\mu} = \mathbb{E}_{p_{\boldsymbol{\theta}}}[\mathbf{t}]$, $\mathbf{u}$ is a unit vector, for $|\gamma| < 1/\beta$.

$$\mathbb{E}[e^{\gamma \mathbf{u}^\top (\mathbf{t} - \boldsymbol{\mu})}] = \int \left(e^{\gamma \mathbf{u}^\top \mathbf{t} - \gamma \mathbf{u}^\top \boldsymbol{\mu}}\right) h(\mathbf{t}) e^{\boldsymbol{\theta}^\top \mathbf{t} - A(\boldsymbol{\theta})} d\mathbf{t}$$

$$= \frac{\exp(-\gamma \mathbf{u}^\top \boldsymbol{\mu})}{Z(\boldsymbol{\theta})} \int h(\mathbf{t}) \exp((\gamma \mathbf{u} + \boldsymbol{\theta})^\top \mathbf{t}) d\mathbf{t}$$

$$= \frac{Z(\gamma \mathbf{u} + \boldsymbol{\theta})}{Z(\boldsymbol{\theta})} \cdot \exp(-\gamma \mathbf{u}^\top \boldsymbol{\mu})$$

The inequality we need to show is equivalent to proving

$$\mathbb{E}[e^{\gamma \mathbf{u}^\top (\mathbf{t} - \boldsymbol{\mu})}] \leq e^{\gamma^2 \nu^2/2}$$

$$\iff \frac{Z(\gamma \mathbf{u} + \boldsymbol{\theta})}{Z(\boldsymbol{\theta})} \cdot e^{-\gamma \mathbf{u}^\top \boldsymbol{\mu}} \leq e^{\gamma^2 \nu^2/2}$$

$$\iff \frac{Z(\gamma \mathbf{u} + \boldsymbol{\theta})}{Z(\boldsymbol{\theta})} \leq e^{\gamma \mathbf{u}^\top \boldsymbol{\mu}} \cdot e^{\gamma^2 \nu^2/2}$$

$$\iff A(\gamma \mathbf{u} + \boldsymbol{\theta}) - A(\boldsymbol{\theta}) \leq \gamma \mathbf{u}^\top \boldsymbol{\mu} + \frac{\gamma^2 \nu^2}{2}$$

Since $A(\boldsymbol{\theta})$ is $L$-smooth, we have that

$$A(\gamma \mathbf{u} + \boldsymbol{\theta}) - A(\boldsymbol{\theta}) \leq \underbrace{\langle \nabla A(\boldsymbol{\theta}), \gamma \mathbf{u} \rangle}_{= \boldsymbol{\mu}} + \frac{L}{2}\|\gamma \mathbf{u}\|^2 = \gamma \mathbf{u}^\top \boldsymbol{\mu} + \frac{\gamma^2 L}{2}$$

where we've used the property of exponential families that the gradient of the log partition function is the mean sufficient statistic. Now we can see that the appropriate parameter for $\nu^2$ is $L$ and $\gamma$ must be small enough so that $\gamma u + \theta \in \Theta$, i.e., $|\gamma| < \frac{1}{\beta}$ for some $\beta > 0$. This is possible if $\theta$ is bounded away from the boundary of $\Theta$. $\square$

*Remark.* In the above, we only needed to use that $p_{\boldsymbol{\theta}}$ is an exponential family distribution and that its log-partition function $A(\boldsymbol{\theta})$ is smooth. It is also possible to show that $p_{\boldsymbol{\theta}}$ has exponentially decreasing tails (in quantities involving $\mathbf{x}$ rather than $T(\mathbf{x})$) if it is log-concave in $\mathbf{x}$ (assumption A2), e.g., by [39].

### C.3   Proof of Lemma 3.4

Let $p_{\boldsymbol{\theta}}(\mathbf{x}) = h(\mathbf{x})\exp(\langle \theta, T(\mathbf{x})\rangle - A(\boldsymbol{\theta}))$ and $A\colon \Theta \to \mathbb{R}$ is the log-partition function:

$$A(\boldsymbol{\theta}) = \int_{\mathcal{X}} h(\mathbf{x})\exp(\langle \theta, T(\mathbf{x})\rangle)d\mathbf{x}.$$

**Lemma C.3.** For any $q > 1$, $\boldsymbol{\theta}, \boldsymbol{\theta}' \in \Theta$:

$$\mathbb{E}_{p_{\boldsymbol{\theta}}}\left[\left(\frac{p_{\boldsymbol{\theta}'}}{p_{\boldsymbol{\theta}}}\right)^q\right] = \exp\left((q-1)A(\boldsymbol{\theta}) - qA(\boldsymbol{\theta}') + A\left(q\boldsymbol{\theta}' - (q-1)\boldsymbol{\theta}\right)\right).$$

*Proof.*

$$\mathbb{E}_{p_{\boldsymbol{\theta}}}\left[\left(\frac{p_{\boldsymbol{\theta}'}}{p_{\boldsymbol{\theta}}}\right)^q\right] = \int_{\mathcal{X}} h(x)\exp\left(\langle \boldsymbol{\theta}, T(x)\rangle - A(\boldsymbol{\theta})\right) \cdot \exp\left(q\langle \boldsymbol{\theta}' - \boldsymbol{\theta}, T(x)\rangle - qA(\boldsymbol{\theta}') + qA(\boldsymbol{\theta})\right) dx$$

$$= \exp\left((q-1)A(\boldsymbol{\theta}) - qA(\boldsymbol{\theta}') + A\left(q\boldsymbol{\theta}' - (q-1)\boldsymbol{\theta}\right)\right).$$

$\square$

**Lemma C.4.** Assume $A$ is convex and $L$-smooth on $\Theta$. For any $S \subseteq \mathcal{X}$, and $\boldsymbol{\theta}, \boldsymbol{\theta}' \in \Theta$:

$$p_{\boldsymbol{\theta}}(S) \geq p_{\boldsymbol{\theta}'}(S)^2 \cdot \exp\left(-\frac{3L}{2}\|\boldsymbol{\theta} - \boldsymbol{\theta}'\|^2\right).$$

*Proof.* By Cauchy-Schwarz,

$$p_{\boldsymbol{\theta}'}(S)^2 = \mathbb{E}_{p_{\boldsymbol{\theta}}}\left[\frac{p_{\boldsymbol{\theta}'}}{p_{\boldsymbol{\theta}}}\mathbf{1}_S\right]^2$$

$$\leq p_{\boldsymbol{\theta}}(S) \cdot \mathbb{E}_{p_{\boldsymbol{\theta}}}\left[\left(\frac{p_{\boldsymbol{\theta}'}}{p_{\boldsymbol{\theta}}}\right)^2\right]$$

$$= p_{\boldsymbol{\theta}}(S) \cdot \exp\left(A(\boldsymbol{\theta}) - 2A(\boldsymbol{\theta}') + A\left(2\boldsymbol{\theta}' - \boldsymbol{\theta}\right)\right).$$

Since $A$ is convex and $L$-smooth,

$$A(\boldsymbol{\theta}) \leq A(\boldsymbol{\theta}') + \langle \nabla A(\boldsymbol{\theta}), \boldsymbol{\theta} - \boldsymbol{\theta}'\rangle$$

$$A(2\boldsymbol{\theta}' - \boldsymbol{\theta}) \leq A(\boldsymbol{\theta}') + \langle \nabla A(\boldsymbol{\theta}'), \boldsymbol{\theta}' - \boldsymbol{\theta}\rangle + \frac{L}{2}\|\boldsymbol{\theta}' - \boldsymbol{\theta}\|^2$$

Therefore,

$$A(\boldsymbol{\theta}) - 2A(\boldsymbol{\theta}') + A\left(2\boldsymbol{\theta}' - \boldsymbol{\theta}\right) \leq \langle \nabla A(\boldsymbol{\theta}') - \nabla A(\boldsymbol{\theta}), \boldsymbol{\theta}' - \boldsymbol{\theta}\rangle + \frac{L}{2}\|\boldsymbol{\theta}' - \boldsymbol{\theta}\|^2$$

$$\leq \frac{3L}{2}\|\boldsymbol{\theta}' - \boldsymbol{\theta}\|^2.$$

$\square$

Compare this to the Gaussian case (e.g., see H.8 of [37]) where this was $p_{\boldsymbol{\theta}}(S) \geq \frac{\alpha}{2}\exp\left(-r \cdot \sqrt{2\log 1/\alpha} - \frac{1}{2}r^2\right)$ for $\|\boldsymbol{\theta} - \boldsymbol{\theta}'\| < r$.

### C.4 Proof of Lemma 3.7

Let $\overline{T} = \frac{1}{n}\sum_{i=1}^{n} T(\mathbf{x}_i)$ be the empirical mean sufficient statistics given our samples $\{\mathbf{x}_i\}_{i=1}^{n}$ each $\mathbf{x}_i \sim p_{\boldsymbol{\theta}^*}^S$.

Let $\epsilon_S > 0$. For $n \geq \Omega\left(\frac{2\beta}{\epsilon_S}\log\left(\frac{1}{\delta}\right)\right)$,

$$\|\overline{T} - \mathbb{E}_{p_{\boldsymbol{\theta}^*}}[T(\mathbf{x})]\| \leq \epsilon_S + \mathcal{O}(\log 1/\alpha)$$

with probability at least $1 - \delta$.

*Proof.* Let $\boldsymbol{\nu}^* = \mathbb{E}_{\mathbf{x} \sim p_{\boldsymbol{\theta}^*}}[T(\mathbf{x})]$.

For any event $A$, we have that

$$\mathbb{P}_{p_{\boldsymbol{\theta}^*}^S}[A] = \int \mathbb{1}\{\omega \in A\}dp_{\boldsymbol{\theta}^*}^S(\omega) = \frac{1}{\alpha}\int \mathbb{1}\{\omega \in A\}\mathbb{1}\{\omega \in S\}dp_{\boldsymbol{\theta}^*}(\omega) \leq \frac{1}{\alpha}\mathbb{P}_{p_{\boldsymbol{\theta}^*}}[A]$$

and for the product measure with $n$ independent components $\mathbb{P}_{\Pi_{i\in[n]}p_{\boldsymbol{\theta}^*}^S}[A] \leq \left(\frac{1}{\alpha}\right)^n \mathbb{P}_{\Pi_{i\in[n]}p_{\boldsymbol{\theta}^*}}[A]$.
So we can bound the probability of events on $p_{\boldsymbol{\theta}^*}^S$ with those on $p_{\boldsymbol{\theta}^*}$. In particular, by Claim 1 and by the composition property of independent sub-exponential random variables, we have that

$$\mathbb{P}_{p_{\boldsymbol{\theta}^*}}\left(\frac{1}{n}\left|\mathbf{u}^\top\left(\sum_i T(\mathbf{x}_i) - \boldsymbol{\nu}^*\right)\right| \geq t\right) \leq \exp\left(-\frac{nt}{2\beta}\right) \qquad \text{for any unit vector } \mathbf{u}$$

$$\Rightarrow \mathbb{P}_{p_{\boldsymbol{\theta}^*}}\left(\left\|\frac{1}{n}\sum_i T(\mathbf{x}_i) - \boldsymbol{\nu}^*\right\| \geq t\right) \leq \exp\left(-\frac{nt}{2\beta}\right).$$

To translate this to the probability of the same event on $p_{\boldsymbol{\theta}^*}^S$, note that

$$\left(\frac{1}{\alpha}\right)^n \exp\left(-\frac{nt}{2\beta}\right) \le \delta \iff \exp\left(n \cdot \left(\log 1/\alpha - \frac{t}{2\beta}\right)\right) \le \delta$$

which holds when $t = 2\beta\left(\log 1/\alpha + \frac{1}{n}\log 1/\delta\right)$. Thus for $n > \frac{2\beta}{\epsilon_S}\log 1/\delta$ samples from the truncated $p_{\boldsymbol{\theta}^*}^S$ we have that with probability at least $1-\delta$, the quantity $\|\overline{T} - \boldsymbol{\nu}^*\| \le 2\beta(\log 1/\alpha) + \epsilon_S$.

$\square$

# D   Additional Proofs for Algorithm Analysis

---
**Algorithm 3** Stochastic Gradient Descent

---
Initialize some $\boldsymbol{\theta}_0 \in K$.
**for** iteration $t = 1, 2, \ldots, T$ **do**
 Compute $\mathbf{v}_t$ such that $\mathbb{E}[\mathbf{v}_t \mid \boldsymbol{\theta}_t] = \nabla f(\boldsymbol{\theta}_t)$
 $\widetilde{\boldsymbol{\theta}}_{t+1} \leftarrow \boldsymbol{\theta}_t - \eta \mathbf{v}_t$
 $\widetilde{\boldsymbol{\theta}}_{t+1} = \Pi_K(\widetilde{\boldsymbol{\theta}}_{t+1})$      (Project onto $K$)
**end for**
Return $\boldsymbol{\theta}_T$

---

## D.1   SGD Algorithm and its Analysis

Although the setting of Theorem 5.7 of [20] is when the objective is a sum of many functions, the proof and its result can be easily adapted to our setting.

**Theorem.**   Let $f$ be a $\lambda$-strongly convex function. Let $\boldsymbol{\theta}^* \in \arg\min_{\boldsymbol{\theta} \in K} f(\boldsymbol{\theta})$. Consider the sequence $\{\boldsymbol{\theta}_t\}_{t=1}^N$ generated by SGD (Algorithm 3) and $\{\mathbf{v}_t\}_{t=1}^N$ the sequence random vectors satisfying $\mathbb{E}[\mathbf{v}_t \mid \boldsymbol{\theta}_t] = \nabla f(\boldsymbol{\theta}_t)$ and $\mathbb{E}[\|\mathbf{v}_t\|^2 \mid \boldsymbol{\theta}_t] < \rho^2$ for all $t$, with a constant step size $\eta$ satisfying $0 < \eta < \frac{1}{\lambda}$. It follows that for $t \ge 0$,

$$\mathbb{E}\|\boldsymbol{\theta}_t - \boldsymbol{\theta}^*\|^2 \le (1 - 2\eta\lambda)^t \|\boldsymbol{\theta}_0 - \boldsymbol{\theta}^*\|^2 + \frac{\eta}{\lambda}\rho^2.$$

*Proof.*  At any iteration $i$,

$$\widetilde{\boldsymbol{\theta}}_{i+1} = \boldsymbol{\theta}_i - \eta \mathbf{v}_i$$
$$\widetilde{\boldsymbol{\theta}}_{i+1} - \boldsymbol{\theta}^* = \boldsymbol{\theta}_i - \boldsymbol{\theta}^* - \eta\mathbf{v}_i \tag{1}$$
$$\|\widetilde{\boldsymbol{\theta}}_{i+1} - \boldsymbol{\theta}^*\|^2 = \|\boldsymbol{\theta}_i - \boldsymbol{\theta}^*\|^2 - 2\eta\langle \mathbf{v}_i, \boldsymbol{\theta}_i - \boldsymbol{\theta}^*\rangle + \eta^2\|\mathbf{v}_i\|^2$$

where the last line comes from multiplying the line (1) with the transpose of the same equation on either side. After projecting to the set $K$ to obtain $\boldsymbol{\theta}_{i+1} = \arg\min_{\boldsymbol{\theta} \in K} \|\widetilde{\boldsymbol{\theta}}_{i+1} - \boldsymbol{\theta}\|^2$ and given that $\boldsymbol{\theta}^* \in K$, we have that $\|\widetilde{\boldsymbol{\theta}}_{i+1} - \boldsymbol{\theta}^*\|^2 \ge \|\boldsymbol{\theta}_{i+1} - \boldsymbol{\theta}^*\|^2$, so

$$\|\boldsymbol{\theta}_{i+1} - \boldsymbol{\theta}^*\|^2 \le \|\boldsymbol{\theta}_i - \boldsymbol{\theta}^*\|^2 - 2\eta\langle \mathbf{v}_i, \boldsymbol{\theta}_i - \boldsymbol{\theta}^*\rangle + \eta^2\|\mathbf{v}_i\|^2 \tag{2}$$

Fixing $\boldsymbol{\theta}_i$ in the $i^{th}$ iteration and taking the conditional expectation in (2) gives

$$\mathbb{E}[\|\boldsymbol{\theta}_{i+1} - \boldsymbol{\theta}^*\|^2 \mid \boldsymbol{\theta}_i] \le \|\boldsymbol{\theta}_i - \boldsymbol{\theta}^*\|^2 - 2\eta\langle \nabla f(\boldsymbol{\theta}_i), \boldsymbol{\theta}_i - \boldsymbol{\theta}^*\rangle + \eta^2\mathbb{E}[\|\mathbf{v}_t\|^2 \mid \boldsymbol{\theta}_i]$$
$$\le (1 - 2\eta\lambda)\|\boldsymbol{\theta}_i - \boldsymbol{\theta}^*\|^2 + \eta^2\mathbb{E}[\|\mathbf{v}_t\|^2 \mid \boldsymbol{\theta}_i]$$

where the last line is due to strong convexity, $\langle \nabla f(\boldsymbol{\theta}_i), \boldsymbol{\theta}_i - \boldsymbol{\theta}^*\rangle \ge \lambda\|\boldsymbol{\theta}_i - \boldsymbol{\theta}^*\|^2$. By taking iterated expectations and recursively applying the above, we get that

$$\mathbb{E}[\|\boldsymbol{\theta}_T - \boldsymbol{\theta}^*\|^2] \le (1 - 2\eta\lambda)^T\|\boldsymbol{\theta}_0 - \boldsymbol{\theta}^*\|^2 + \eta^2\rho^2 \sum_{i=0}^{T-1}(1 - 2\eta\lambda)^i$$

$$\le (1 - 2\eta\lambda)^T\|\boldsymbol{\theta}_0 - \boldsymbol{\theta}^*\|^2 + \eta^2\frac{1}{\eta\lambda}\rho^2$$

$$= (1 - 2\eta\lambda)^T\|\boldsymbol{\theta}_0 - \boldsymbol{\theta}^*\|^2 + \frac{\eta}{\lambda}\rho^2$$

where in the second line we used that $\sum_{i=0}^{T-1}(1-2\eta\lambda)^i = \frac{1-(1-2\eta\lambda)^T}{1-(1-2\eta\lambda)} < \frac{2}{2\eta\lambda}$ provided $\eta < 1/\lambda$. $\square$

We can derive the complexity (number of iterations) to get $\mathbb{E}[\|\boldsymbol{\theta}_T - \boldsymbol{\theta}\|^2] < \epsilon$ using the following Lemma from [20].

**Lemma D.1** (Lemma A.2 of [20]). Consider the recurrence given by
$$\alpha_k \leq (1-\eta\mu)^t\alpha_0 + A\eta,$$
where $\mu > 0$, and $A, C \geq 0$ are given constants and $\eta < 1/C$. If
$$\eta = \min\left\{\frac{\epsilon}{2A}, \frac{1}{C}\right\}$$
then
$$t \geq \max\left\{\frac{1}{\epsilon}\frac{2A}{\mu}, \frac{C}{\mu}\right\}\log\left(\frac{2\alpha_0}{\epsilon}\right) \Rightarrow \alpha_k \leq \epsilon.$$

Note that to get bounds on $\|\boldsymbol{\theta}_T - \boldsymbol{\theta}^*\|$ rather than $\|\boldsymbol{\theta}_T - \boldsymbol{\theta}^*\|^2$, we can solve the number of iterations we need to get $\epsilon^2$ on the right hand side, and we will get the number of iterations for $\|\boldsymbol{\theta}_T - \boldsymbol{\theta}^*\| < \epsilon$. Then the resulting complexity bounds will replace $1/\epsilon$ with $1/\epsilon^2$.

## D.2 Approximate Sampling of Non-Truncated Distribution

Many analyses of stochastic gradient descent assume unbiased directions at every iteration of the algorithm, but since we need to be able to sample from $p_{\boldsymbol{\theta}_i}$ multiple times at each iteration $i$ until we get a sample in $S$, our directions are only unbiased if we can indeed sample exactly from $p_{\boldsymbol{\theta}_i}$ each time for all $i$.

However if $p_{\boldsymbol{\theta}_i}$ is complicated, exact sampling can be difficult or take too long. Since we assume that $p_{\boldsymbol{\theta}}$ is log-concave in $\mathbf{x}$ for all $\boldsymbol{\theta} \in \Theta$, we can at least approximately sample from it efficiently via Langevin Monte Carlo, MALA, or other algorithms with convergence guarantees for log-concave densities.

**Lemma D.2** (SGD Analysis with Biased Directions). Let $f$ be a $\lambda$-strongly convex function. Let $\boldsymbol{\theta}^* \in \arg\min_{\boldsymbol{\theta}\in K\subseteq B(\boldsymbol{\theta}^*, D)} f(\boldsymbol{\theta})$. Consider the sequence $\{\boldsymbol{\theta}_t\}_{t=1}^N$ generated by SGD but with $\{\widetilde{\mathbf{v}}_t\}_{t=1}^N$ the sequence random vectors satisfying $\mathbb{E}[\widetilde{\mathbf{v}}_t \mid \boldsymbol{\theta}_t] = \mathbf{b}_t - \nabla f(\boldsymbol{\theta}_t)$ with $\|\mathbf{b}_t\| < B$ and $\mathbb{E}[\|\widetilde{\mathbf{v}}_t\|^2 \mid \boldsymbol{\theta}_t] < \rho'^2$ for all $t$, with a constant step size $\eta$ satisfying $0 < \eta < \frac{1}{\lambda}$. It follows that for $t \geq 0$,
$$\mathbb{E}\|\boldsymbol{\theta}_t - \boldsymbol{\theta}^*\|^2 \leq (1-2\eta\lambda)^t\|\boldsymbol{\theta}_0 - \boldsymbol{\theta}^*\|^2 + \frac{\eta}{\lambda}\rho^2 + \frac{2BD}{\lambda}.$$

*Proof.* Define $\mathbf{b}_t := \mathbb{E}[\mathbf{b}_t \mid \boldsymbol{\theta}_t] - \nabla f(\boldsymbol{\theta}_t)$ the bias for each $t \geq 0$. The analysis of Theorem 3.10 can be applied generically to get Eq. (2):
$$\|\boldsymbol{\theta}_{i+1} - \boldsymbol{\theta}^*\|^2 \leq \|\boldsymbol{\theta}_i - \boldsymbol{\theta}^*\|^2 - 2\eta\langle\widetilde{\mathbf{v}}_i, \boldsymbol{\theta}_i - \boldsymbol{\theta}^*\rangle + \eta^2\|\widetilde{\mathbf{v}}_i\|^2$$

Now when taking the conditional expectation, we get
$$\mathbb{E}[\|\boldsymbol{\theta}_{i+1} - \boldsymbol{\theta}^*\|^2 \mid \boldsymbol{\theta}_t] \leq \|\boldsymbol{\theta}_i - \boldsymbol{\theta}^*\|^2 - 2\eta\langle\mathbf{b}_t, \boldsymbol{\theta}_i - \boldsymbol{\theta}^*\rangle - 2\eta\langle\nabla f(\boldsymbol{\theta}_i), \boldsymbol{\theta}_i - \boldsymbol{\theta}^*\rangle - \eta^2\mathbb{E}[\|\widetilde{\mathbf{v}}_i\|^2 \mid \boldsymbol{\theta}_i]$$
$$\leq (1-2\eta\lambda)\|\boldsymbol{\theta}_i - \boldsymbol{\theta}^*\|^2 + \eta^2\rho'^2 + 2\eta\|\mathbf{b}_i\|\|\boldsymbol{\theta}_i - \boldsymbol{\theta}^*\|$$

Taking iterated expectations and recursively applying this gives now
$$\mathbb{E}[\|\boldsymbol{\theta}_T - \boldsymbol{\theta}^*\|^2] \leq (1-2\eta\lambda)^T\|\boldsymbol{\theta}_0 - \boldsymbol{\theta}^*\|^2 + \sum_{i=0}^{T-1}(1-2\eta\lambda)^i \cdot \left(\eta^2\rho'^2 + 2\eta\|\mathbf{b}_i\|\|\boldsymbol{\theta}_i - \boldsymbol{\theta}^*\|\right)$$
$$\leq (1-2\eta\lambda)^T\|\boldsymbol{\theta}_0 - \boldsymbol{\theta}^*\|^2 + \frac{\eta^2\rho'^2 + 2\eta BD}{\eta\lambda}$$
$$= (1-2\eta\lambda)^T\|\boldsymbol{\theta}_0 - \boldsymbol{\theta}^*\|^2 + \frac{\eta\rho'^2}{\lambda} + \frac{2BD}{\lambda}$$
where the second line holds if $\|\mathbf{b}_i\| \leq B, \|\boldsymbol{\theta}_i - \boldsymbol{\theta}^*\| < D, \forall i$ (which holds under the assumptions). $\square$

Note that in our Algorithm 1, $D$ here is simply $\frac{2}{\lambda}d(\alpha)$ by construction. We can also control $B$ through the following.

**Bounding the bias.** Fix some $t$. Let $\widetilde{\mathbf{v}}_t := T(\mathbf{z}) - T(\mathbf{x})$ where $\mathbf{z} \sim \widetilde{p}_{\boldsymbol{\theta}_i}^S$ and $\mathbf{x} \sim p_{\boldsymbol{\theta}^*}^S$. Then we can write

$$
\begin{aligned}
\mathbf{b}_t &= \mathbb{E}_{\mathbf{z} \sim \widetilde{p}_{\boldsymbol{\theta}_i}^S}[T(\mathbf{z})] - \mathbb{E}_{\mathbf{x} \sim p_{\boldsymbol{\theta}^*}^S}[T(\mathbf{x})] + \mathbb{E}_{\mathbf{z} \sim p_{\boldsymbol{\theta}_i}^S}[T(\mathbf{z})] - \mathbb{E}_{\mathbf{x} \sim p_{\boldsymbol{\theta}^*}^S}[T(\mathbf{x})] \\
&= \mathbb{E}_{\mathbf{z} \sim \widetilde{p}_{\boldsymbol{\theta}_i}^S}[T(\mathbf{z})] - \mathbb{E}_{\mathbf{z} \sim p_{\boldsymbol{\theta}_i}^S}[T(\mathbf{z})] \\
&= \int_{\mathcal{X}} T(\mathbf{x}) \cdot (\widetilde{p}_{\boldsymbol{\theta}_i}^S(\mathbf{x}) - p_{\boldsymbol{\theta}_i}^S(\mathbf{x})) d\mathbf{x}
\end{aligned}
\tag{3}
$$

If we know that $T(\mathbf{x})$ is bounded over $S$, we can upper-bound this given the TV distance between $\widetilde{p}_{\boldsymbol{\theta}_i}^S$ and $p_{\boldsymbol{\theta}_i}^S$:

$$
\|\mathbf{b}_t\| \leq \sup_{\mathbf{x} \in S} \|T(\mathbf{x})\| \left\| \widetilde{p}_{\boldsymbol{\theta}_i}^S - p_{\boldsymbol{\theta}_i}^S \right\|_{TV}.
$$

Since we assume that $\mathbb{E}_{p_{\boldsymbol{\theta}_i}}[T(\mathbf{x})]$ is finite, it should be the case that $T(\mathbf{x})$ is bounded over its support except potentially on some negligible sets. In that case, we can replace $T(\mathbf{x})$ with $\widetilde{T}(\mathbf{x})$ which replaces those potentially infinite values on negligible sets with 0 and the integral expression in (3) would be equal to one which uses $\widetilde{T}(\mathbf{x})$ instead of $T(\mathbf{x})$, and the bound on its norm holds given that $\widetilde{T}(\mathbf{x})$ is bounded.

Otherwise we can use bounds from Lemma C.1 to bound this as

$$
\|\mathbf{b}_t\| \leq \sqrt{\chi^2(\widetilde{p}_{\boldsymbol{\theta}_i}^S \| p_{\boldsymbol{\theta}_i}^S)} \cdot \sqrt{\mathbf{Var}_{p_{\boldsymbol{\theta}_i}^S}(T(\mathbf{x}))}
$$

if we have control over the chi-square divergence (see Section C.1 for definitions).

If we know that $T(x)$ is a 1-Lipschitz, real-valued function (e.g., when $T(x) = x$), we can use the dual representation of $W_1$ distance to bound this as

$$
\mathbf{b}_t = \int_S T(x) d\widetilde{p}_{\boldsymbol{\theta}_i}^S - \int_S T(x) dp_{\boldsymbol{\theta}_i}^S \leq \sup_{f \in \mathcal{F}_{1\mathrm{Lip}}} \int f(x) d\widetilde{p}_{\boldsymbol{\theta}_i}^S - \int f(x) dp_{\boldsymbol{\theta}_i}^S = W_1(\widetilde{p}_{\boldsymbol{\theta}_i}^S, p_{\boldsymbol{\theta}_i}^S).
$$

**Proposition D.3** (Bounds on truncated total variation, given bounds on non-truncated). Suppose $\|\widetilde{p}_{\boldsymbol{\theta}_i} - p_{\boldsymbol{\theta}_i}\|_{TV} \leq \epsilon_{TV}$ for some $\epsilon_{TV} > 0$. Then

$$
\|\widetilde{p}_{\boldsymbol{\theta}_i}^S - p_{\boldsymbol{\theta}_i}^S\|_{TV} \leq \frac{\epsilon_{TV}^2}{p_{\boldsymbol{\theta}_i}(S) - \epsilon_{TV}}.
$$

*Proof.* First, given that $\|\widetilde{p}_{\boldsymbol{\theta}_i} - p_{\boldsymbol{\theta}_i}\|_{TV} \leq \epsilon_{TV}$, we have by one characterization of the total variation distance (the supremum of the difference in mass over all measurable sets)

$$
\widetilde{p}_{\boldsymbol{\theta}_i}(S) \geq p_{\boldsymbol{\theta}_i}(S) - \epsilon_{TV}.
$$

Now for the truncated densities,

$$
\begin{aligned}
\|\widetilde{p}_{\boldsymbol{\theta}_i}^S - p_{\boldsymbol{\theta}_i}^S\|_{TV} &= \frac{1}{2} \int |\widetilde{p}_{\boldsymbol{\theta}_i}^S(\mathbf{x}) - p_{\boldsymbol{\theta}_i}^s(\mathbf{x})| d\mathbf{x} \\
&= \frac{1}{2} \int \mathbb{1}\{\mathbf{x} \in S\} \cdot \left| \frac{\widetilde{p}_{\boldsymbol{\theta}_i}(\mathbf{x})}{\widetilde{p}_{\boldsymbol{\theta}_i}(S)} - \frac{p_{\boldsymbol{\theta}_i}(\mathbf{x})}{p_{\boldsymbol{\theta}_i}(S)} \right| d\mathbf{x} \\
&\leq \frac{\epsilon_{TV}}{p_{\boldsymbol{\theta}_i}(S) - \epsilon_{TV}} \cdot \frac{1}{2} \int |\widetilde{p}_{\boldsymbol{\theta}_i}(\mathbf{x}) - p_{\boldsymbol{\theta}_i}(\mathbf{x})| d\mathbf{x} \\
&\leq \frac{\epsilon_{TV}^2}{p_{\boldsymbol{\theta}_i}(S) - \epsilon_{TV}}
\end{aligned}
$$

$\square$

**Efficient, Approximate Sampling.** There exist several results in sampling which give bounds in TV distance in polynomial time (mixing time bounds) for log-concave distributions, e.g., [15], [3], [33] but which also usually require that the log density is also smooth (in $\mathbf{x}$, not $\boldsymbol{\theta}$). There are also proofs for Langevin Monte Carlo when the log density is convex and Lipschitz, not necessarily smooth (e.g., see Chapter 4 of [7]), or under LSI (which is implied by strong log-concavity) with convergence in Renyi divergence (e.g., Chapter 5 of [7]). We can also use the proximal sampler to achieve convergence in KL divergence under log-concavity (e.g., Chapter 8.4 of [7]), which by Pinsker's inequality can bound the TV distance.

**Proposition D.4** (Bounded variance step with bias). If $\mathbb{E}[\|\mathbf{v}_i\| \mid \boldsymbol{\theta}_i] \leq \rho^2$ where $\mathbf{v}_i = T(\mathbf{z}) - T(\mathbf{x})$ with $\mathbf{z} \sim p_{p^S_{\boldsymbol{\theta}_i}}$ and $\mathbf{x} \sim p^S_{\boldsymbol{\theta}^*}$, then $\mathbb{E}[\|\widetilde{\mathbf{v}}_i\| \mid \boldsymbol{\theta}_i] \leq \rho'^2$ for $\widetilde{\mathbf{v}}_i = T(\widetilde{\mathbf{z}}) - T(\mathbf{x})$ with $\widetilde{\mathbf{z}} \sim \widetilde{p}_{p^S_{\boldsymbol{\theta}_i}}$ and $\mathbf{x} \sim p^S_{\boldsymbol{\theta}^*}$, where

$$\rho'^2 = \mathbf{Var}_{\widetilde{p}^S_{\boldsymbol{\theta}_i}}(T(\mathbf{z})) - \mathbf{Var}_{p^S_{\boldsymbol{\theta}_i}}(T(\mathbf{z})) + \rho^2 + B^2,$$

where $\|\mathbf{b}_i\| = \|\mathbb{E}_{\widetilde{\mathbf{z}} \sim \widetilde{p}^S_{\boldsymbol{\theta}_i}}[T(\widetilde{\mathbf{z}})] - \mathbb{E}_{\mathbf{z} \sim p^S_{\boldsymbol{\theta}_i}}[T(\mathbf{z})]\| < B$.

*Proof.* As in the proof of the exact sampling version, we can write

$$
\begin{aligned}
\mathbb{E}[\|\widetilde{\mathbf{v}}_i\| \mid \boldsymbol{\theta}_i] &= \mathbf{Var}_{\widetilde{p}^S_{\boldsymbol{\theta}_i}}(T(\mathbf{z})) + \mathbf{Var}_{p^S_{\boldsymbol{\theta}^*}}(T(\mathbf{x})) + \|\mathbb{E}_{\widetilde{p}^S_{\boldsymbol{\theta}_i}}[T(\mathbf{z})] - \mathbb{E}_{p^S_{\boldsymbol{\theta}^*}}[T(\mathbf{x})]\|^2 \\
&\leq \mathbf{Var}_{\widetilde{p}^S_{\boldsymbol{\theta}_i}}(T(\mathbf{z})) + \mathbf{Var}_{p^S_{\boldsymbol{\theta}^*}}(T(\mathbf{x})) + \|\mathbb{E}_{p^S_{\boldsymbol{\theta}_i}}[T(\mathbf{z})] - \mathbb{E}_{p^S_{\boldsymbol{\theta}^*}}[T(\mathbf{x})]\|^2 + \|\mathbf{b}_i\|^2 \\
&= \mathbf{Var}_{\widetilde{p}^S_{\boldsymbol{\theta}_i}}(T(\mathbf{z})) - \mathbf{Var}_{p^S_{\boldsymbol{\theta}_i}}(T(\mathbf{z})) \\
&\quad + \underbrace{\mathbf{Var}_{p^S_{\boldsymbol{\theta}_i}}(T(\mathbf{z})) + \mathbf{Var}_{p^S_{\boldsymbol{\theta}^*}}(T(\mathbf{x})) + \|\mathbb{E}_{p^S_{\boldsymbol{\theta}_i}}[T(\mathbf{z})] - \mathbb{E}_{p^S_{\boldsymbol{\theta}^*}}[T(\mathbf{x})]\|^2}_{\leq \rho^2} + \underbrace{\|\mathbf{b}_i\|^2}_{\leq B^2} \\
&\leq \mathbf{Var}_{\widetilde{p}^S_{\boldsymbol{\theta}_i}}(T(\mathbf{z})) - \mathbf{Var}_{p^S_{\boldsymbol{\theta}_i}}(T(\mathbf{z})) + \rho^2 + B^2
\end{aligned}
$$

$\square$

We can bound the difference $\mathbf{Var}_{\widetilde{p}^S_{\boldsymbol{\theta}_i}}(T(\mathbf{z})) - \mathbf{Var}_{p^S_{\boldsymbol{\theta}_i}}(T(\mathbf{z}))$ if have bounds on $\mathbf{Var}_{\widetilde{p}^S_{\boldsymbol{\theta}_i}}(T(\mathbf{z}))$, through bounds like in Lemma C.1 if we can say something about the chi-square divergence, or through similar arguments to the bias bound if we assume some bounds on $\|T(\mathbf{x})\|^2$ over its support.

# E   Numerical Example

To illustrate how the algorithm performs in different dimensions, we implemented our algorithm for 2-, 5-, 10-, and 20-dimensional exponential distributions. In all cases, the truncation set is the (hyper-)cube $[0, 2]^d$. We chose true parameters in all cases which resulted in an initial error at most 2.5. In all cases, we use 1500 iterations and step size 0.01, each repeated 10 times. In the end, all have (average) L2 error at most 0.15. For stability (and to bypass repeating the algorithm multiple times as stated in the analysis), we instead calculated gradients using the average of 10 samples which was sufficient to have stable training results. See Figure 2.

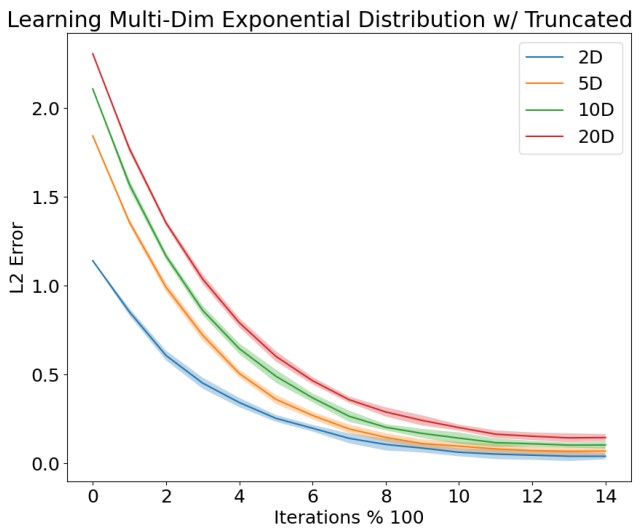

Figure 2: Learning 2-, 5-, 10-, and 20-dimensional truncated exponential distributions. In all cases, the truncation set is the (hyper-)cube $[0, 2]^d$.

The wall clock time to finish all 1500 iterations of training for 5-, 10-, and 20-dimensions was $42.9 \pm 2.2$, $49.1 \pm 6.5$, and $61.2 \pm 3.0$ seconds, respectively. We can see that the running time is not doubling with the doubling of dimensions.

