# OpenReview forum: "Learning Exponential Families from Truncated Samples"
_NeurIPS.cc/2023/Conference — NeurIPS 2023 poster_

### Official Review · Reviewer_SUu1 · 2023-06-22

**Soundness:** 4 excellent
**Presentation:** 2 fair
**Contribution:** 2 fair
**Rating:** 6
**Confidence:** 3

**Summary:**

This paper considers a case where one has observations from a censored exponential family (i.e., an exponential family, but each distribution is conditioned to lie within a particular set) and wants to estimate the parameters of that exponential family.  An additional complication in the setup is that one does not have access to the set that observations are conditioned to lie within, but instead one has access to an oracle that can by queried and will tell you whether any given point lies within the truncation set or not.  This exact setting has been considered previously but restricted to less general families of distributions (e.g., multivariate Gaussians).  The authors develop a provably accurate estimator for the parameters of the exponential family that runs in polynomial time in all of the relevant parameters.  The main idea is to first perform maximum likelihood as if the data were not censored, and then perform projected stochastic gradient descent.  Stochastic gradient estimates can be obtained in this setting by using the oracle to perform rejection sampling using the non-truncated distribution defined by the current parameter estimate as a proposal distribution.

**Strengths:**

* The proofs are thorough, and, as far as I can tell, correct.
* The authors make an admirable attempt to first discuss their assumptions and results at a non-technical level before present the more technical formal results.
* The results are quite general, and the proposed algorithm is exceptionally simple and straightforward, making the approach appealing and elegant.

**Weaknesses:**

* The flow of the technical part of the paper was very difficult to follow.  The major presentation aspect that I felt was missing was a big picture overview of both the algorithm and the proof technique.  I felt like there was too large a disconnect between the informal statement of the results, and then diving directly into technical assumptions and lemmas without giving any intuition about why they were important or where they would be used.  I realize that space is limited in this venue, but perhaps more of the technical results could be moved to the supplement to make more room.
* I also feel that the paper is missing some motivation.  To me, generalizing existing results is not in itself interesting to a broad audience.  I could imagine that the generalization presented in this paper is broadly useful, but some additional context would go a long way toward proving that point.  In particular, I am having trouble imagining a scenario where one has access to an oracle that can tell whether a given point would have been censored or not, but one does not have access to knowledge of the censoring set itself.  Just listing some specific, motivating real world examples would be useful.  I am not embedded in the truncated statistics community, so perhaps such motivation is obvious, but to a broader community I think additional motivation would be useful.
* Lastly, some numerical examples checking the proof results would be useful.  How does running projected SGD on truncated data perform in some real world cases?  A comprehensive application or simulation study is obviously beyond the scope of this paper (and not necessary) but just a few toy examples would go a long way in giving an independent check of the analytical results.

Typos / minor comments:
* Line 34: "is the first to provide" --> "are the first to provide"
* Line 76: Totally up to the authors, but it may be helpful to remind the reader that $m$ is the dimension of $\theta$ here.
* Line 192: "log likelihood" --> "log-likelihood"
* Line 271: this sentence is not grammatically correct
* Equation following Line 272: The "$B$" notation has not been defined.  Presumably it's an ($\ell_2$?) ball centered at the first argument with radius the second argument.
* In the equations around lines 458-460, it would be good to use a different dummy variable than $\mathbf{x}$ in the integral as $\mathbf{x}$ is already being used outside of the integral.
* Equation after line 469: $p_\xi^*$ is not defined, but based on the argument it matches the definition of $p_\xi$ in the text above line 469.
* Line 575: I believe that instead of $\rho^S$ it should be $\pi$.

**Questions:**

* The wording of assumption II is ambiguous.  Do the exponential families need to contain _exclusively_ log-concave distributions?  Seems like yes from Assumption A2.

* Does assumption A4 require additional assumptions on $g(s)$ (e.g., being finite)?  It seems like one could _define_ g(s) as $\mathbb{E}[X | X \ge s]$ and then the condition vacuously holds.

* The informal version of the main result has a dependence on sample size, but the formal version does not.  I believe that the dependence on $n$ comes from $\epsilon$ but in the setup of the theorem, it says we are given samples $\mathbf{x}_1, \ldots, \mathbf{x}_n$ which makes $n$ seem fixed.

**Limitations:**

The authors do not include a broader impact statement.  I don't believe the work has any direct negative societal impacts, but it would be good for the authors to say so.  Additionally, I could imagine that estimating population parameters from censored data could, in principle, pose privacy risks, and so the authors may wish to make some minor comments along these lines.

---

> ### Author Rebuttal · Authors · 2023-08-09
>
> We would like to thank the reviewer for a thorough and careful treatment in reviewing our work! We will do our best to address all of the weaknesses and questions listed in your review.
>
> ### **Weaknesses:**
>
> 1. _"The flow of the technical part of the paper was very difficult to follow... I realize that space is limited in this venue, but perhaps more of the technical results could be moved to the supplement to make more room."_
>
> **Answer:** We apologize if the paper was not written clearly enough initially – as we’ve mentioned in our other responses, we will make efforts to reorganize the presentation of the technical results specifically by (1) moving some less essential lemmas to the Appendix and (2) moving Section 3.3 after 3.1, followed by 3.2, explaining why we need these lemmas along the way (e.g., Section 3.1 establishes that the strong convexity and smoothness parameters depend on $p_\theta(S)$, which we bound in Section 3.3 as a function of parameter distances, after which 3.2 provides a good initialization). We are happy to hear any other suggestions.
>
> 2. _"I also feel that the paper is missing some motivation... In particular, I am having trouble imagining a scenario where one has access to an oracle that can tell whether a given point would have been censored or not... Just listing some specific, motivating real world examples would be useful..."_
>
> **Answer:** Thank you for the feedback! We agree that it would strengthen the paper to better present the motivation. We had focused a lot of efforts in just presenting the main result.
> * To the point about generalizing results, while estimation from truncated data is a very classical problem with a long history in statistics, only recently did we have efficient algorithms to solve the problem in high dimensions for Gaussian distributions. Since many practical manifestations of missing data problems may not always have Gaussianity in the data generating distribution or the noise, any generalization gives the user more degrees of freedom to model real problems. It would be great to understand more completely how to solve these problems for even more general distributions beyond exponential families, but even this extension beyond Gaussians is quite new (and all prior results use properties of Gaussian distributions).
> * We also provide some more motivation/examples in the general response to all reviewers (please see above).
>
> 3. _"Lastly, some numerical examples checking the proof results would be useful..."_
>
> **Answer:** As mentioned in our responses to the other reviewers, we did not include numerical demonstrations due to (1) the theoretical results already taking a lot of space to present (including the appendices), (2) it is not usually included in similar/related works. It is also our opinion that while simple simulated experiments may be less helpful for this paper than using the limited space for more intuition and exposition, real-world application may be beyond the scope of the current work.
>
> ### **Questions:**
>
> 1. _"The wording of assumption II is ambiguous. Do the exponential families need to contain exclusively log-concave distributions? Seems like yes from Assumption A2."_
>
> **Answer:** Yes, the exponential families (before truncation) need to contain exclusively log-concave distributions. We will rewrite the assumption to make it more clear (perhaps instead of writing $p_\theta$ we will write “the exponential family distributions we consider before truncation are log-concave”).
>
> 2. _"Does assumption A4 require additional assumptions on $g(s)$ (e.g., being finite)? It seems like one could define $g(s)$ as $E[X \mid X \geq s]$ and then the condition vacuously holds."_
>
> **Answer:** Thank you for catching this, and we apologize that the wording was ambiguous. But yes we need $g(s)$ to be a uniform upper bound on $E[X | X \geq s]$ for all distributions in the exponential family that we consider and should be finite for finite $s$, i.e., it should be that for every finite $s$: $\sup_{p \in \mathcal{P}} E_p[X | X \geq s] \leq g(s) < \infty$ if $\mathcal{P}$ is the class of exponential families we consider. The second important property of $g(s)$ is that it should not increase as the dimension increases. We commit to rephrasing this assumption to clarify this in the final version of the paper.
>
> 3. _"The informal version of the main result has a dependence on sample size, but the formal version does not. I believe that the dependence on $n$ comes from $\epsilon$ but in the setup of the theorem, it says we are given samples $x_1,…,x_n$ which makes $n$ seem fixed."_
>
> **Answer:** Sorry for the confusion here. Once we choose an $\epsilon$, this determines how many samples ($O(k/\epsilon^2)$) we will need (and iterations) to get the $\epsilon$ error in parameters. The informal version simply wrote it the other way: given $n$, the error in parameters is $O(\sqrt{k/n})$ (solving for $n = k/\epsilon^2$). The informal one is missing some technical details, for instance $n$ needs to be at least a certain value for this to make sense. Thank you for catching this and we will update the informal version to match the notation of our main theorem.
>
> ### **Other:**
>
> 1. _"The authors do not include a broader impact statement. I don't believe the work has any direct negative societal impacts, but it would be good for the authors to say so. Additionally, I could imagine that estimating population parameters from censored data could, in principle, pose privacy risks, and so the authors may wish to make some minor comments along these lines."_
>
> **Answer:** This is a good point! Our results indeed illustrate that just hiding some part of the space might not suffice to guarantee the privacy of that space. We commit to adding this in the final version of this manuscript.
>
> 2. Typos
>
> **Answer:** Thank you so much for your careful reading! We will fix the various typos you have kindly taken the time to point out. Thank you again.

---

> > ### Comment · Reviewer_SUu1 · 2023-08-10
> >
> > Thank you for the thorough and clear response -- it cleared up my confusion about the informal assumptions.  I also appreciate the willingness to improve the presentation of the paper.  The motivation is also much clearer now.  For me, the most compelling motivation is when the truncation set is known but difficult to deal with, but it is easier to decide whether a given point is inside or outside -- thank you for the very clear motivation.
> >
> > I do disagree, however, with using the space constraint as a reason to avoid a (very small, even toy) empirical study.  In the main text, there could be a single sentence such as "See Appendix ?? for an empirical example of the proposed algorithm," with all of the remaining details in the appendix.  While I appreciate that some previous work did not include empirical checks, I also do not see that as a good reason to avoid doing them now.  One of the beautiful aspects of this work to me is its simplicity, and having a simple implementation and applying it to a simple problem would highlight this simplicity (as well as providing an independent check on its utility/correctness).
> >
> > Again, thank you for the response, and I hope that the comments were helpful.  In light of the response, I will increased my score to "weak accept".

---

> > > ### Author Response · Authors · 2023-08-10
> > >
> > > Thank you so much for reading our response and following up so quickly! If the reviewers believe that this a crucial addition to our paper, we can commit to including some (perhaps simple, toy) numerical demonstration in the final version of the paper. If the other content takes a lot of space in the main body, having the experiments in the Appendix is a good compromise -- thank you for the suggestion!

---

### Official Review · Reviewer_UJWe · 2023-06-30

**Soundness:** 3 good
**Presentation:** 3 good
**Contribution:** 3 good
**Rating:** 6
**Confidence:** 3

**Summary:**

This paper focuses on estimating a parametric family of distributions using a truncated sample, i.e.  the sample can only be collected within a subset of the data domain. The primary objective is to develop efficient algorithms for estimating the parameters of this family. To achieve this, previous findings on truncated samples from the Gaussian case are extended to exponential families.

**Strengths:**


This is a theoretical paper that offers valuable support for the proposed algorithms. The extension to exponential families is significant.


**Weaknesses:**

This paper is solely theoretical and lacks numerical support for the proposed algorithms and theorems. It does not provide concrete examples to motivate the study. For instance, it would be helpful to provide real-world examples where the oracle property holds.





**Questions:**

You mentioned that "Some intermediate results can even be generalized to distributions beyond exponential families."  What are the key factors that are needed for such an extension?  Can you provide some concrete examples where such extensions are feasible?


**Limitations:**

I haven’t come across any stated limitations, but it’s possible that I may have overlooked them.

---

> ### Author Rebuttal · Authors · 2023-08-09
>
> We thank the reviewer for the feedback! We will take care to use this space to address your main concerns and questions:
>
> 1. _"This paper is solely theoretical and lacks numerical support for the proposed algorithms and theorems. It does not provide concrete examples to motivate the study. For instance, it would be helpful to provide real-world examples where the oracle property holds."_
>
> **Answer:**
>  * Thank you for your feedback. As we also mentioned in our other responses, we did not include numerical demonstrations due to (1) the theoretical results already taking a lot of space to present (including the appendices), (2) it is not usually included in similar/related works.
>
> * The membership oracle assumption has been shown to be necessary to solve this problem even in the Gaussian case (the previous work shows an impossibility result without even membership oracle access to the set S). We will make this more clear and add more explanation after this assumption. Practically, the membership oracle is an abstract but weak access to the set S; even if we know what S is, in practice it can be difficult to (mathematically) describe it yet much easier to decide whether a point is inside it or not.
>
> * In the response to all reviewers, we also give more concrete examples (please see above).
>
> 2. _"You mentioned that "Some intermediate results can even be generalized to distributions beyond exponential families." What are the key factors that are needed for such an extension? Can you provide some concrete examples where such extensions are feasible?"_
>
> **Answer:**
>
> * We apologize if the wording was not very clear. Some results like that of Lemma 3.4 (and Appendix C.1) hold very generally for any distribution over $\mathbb{R}^m$. For instance, the KL divergence between (any) density and its truncation is $\log 1 / \alpha$, if $\alpha$ is the mass of the truncation set under the density. We hope that these kinds of intermediate results can be used in the future to solve more general problems.
>
> * While we do have some of these lemmas which hold in general cases, to have a full comprehensive analysis we had to use properties of exponential families (e.g., Section 3.1, Lemma 3.7, Lemma 3.9). We leave for future work understanding precisely the limits of how much more this framework can be extended to general densities.

---

> > ### Comment · Reviewer_UJWe · 2023-08-12
> >
> > I appreciate your response indicating that the theoretical results have already occupied a lot of space, and that previous related works did not include numerical demonstrations.  However, I believe it is important to provide numerical support for a theoretical paper, which could be included in the supplementary materials. In the main paper, a brief mention of real-world examples or references would increase the significance and relevance of the problem.

---

> > > ### Author Response · Authors · 2023-08-12
> > >
> > > Thank you so much for your response! If the reviewers agree that it would be beneficial to add (even small) numerical demonstrations in the supplementary, we can commit to including them in the final version. We can also include some of the real-world examples (and related references) that we listed in the general response to all reviewers in the main paper. Thank you for your feedback!

---

> > > > ### Comment · Reviewer_UJWe · 2023-08-12
> > > >
> > > > I have increased my rating to 6 (weak accept) assuming that you will add numerical demonstrations in the supplementary materials.

---

### Official Review · Reviewer_P6iL · 2023-07-05

**Soundness:** 3 good
**Presentation:** 3 good
**Contribution:** 4 excellent
**Rating:** 6
**Confidence:** 3

**Summary:**

The paper proposes a new general method for estimating the parameters of exponential-family (EF) distributions from truncated samples. The method is presented as an algorithm, which allow us to obtain good estimates of $\theta$ in polynomial time and certain accuracy conditions. The whole methodology extends previous advances in parameter estimation from truncated samples (these last ones mainly with Gaussian densities and regression problems). The formal results included seem to be theoretically well grounded on the properties of exponential families, exploiting previous methods to both lower and upper bound the covariance of the sufficient statistics given the truncated samples.

In practice, the algorithm looks efficient (to me), as it seems to compute a certain gradient step given a measure of distance between the sufficient statistic functions of the truncated samples and the true samples of the formal EF distribution. Later, this parameter update is corrected (as it was only computed from a truncated sample that might guide it towards bad locations), and in particular, projected into the "correct" parameter space where it will finally converge to the desired estimate.

**Strengths:**

Despite some points and details that I will later add, I liked the paper quite a lot, in my opinion, the construction of the theory around exponential families is clear and strong. This last point is indeed not easy, as EF have very good properties that are however not that easy to exploit without the proper experience. In that sense, I also found very strong the use of such properties to build the main properties of the objective in section 3.1 (the three equations behind the first paragraph) which to me seem to open the door to the main result provided in the algorithm. Also surprisingly, the algorithm for estimation seems easy and direct to use, even if theory is a bit more complex. To me the paper has several strengths that is worth remarking: 1) the generalisation of the previous theory on estimation from truncated samples to exponential families, 2) the clarity and utility of the exponential family principles to make the parameter estimation possible and 3) the final result provided in the algorithm which makes the estimation possible.

**Weaknesses:**

In my opinion, there are two lingering points that makes me think the paper is perhaps weak in some degree..

**Point 1.** First, even if I liked the idea, the utility, the algorithm and the results provided in a significant way, I had a hard time reading it and kind of building myself the answers to what the main contributions and strengths of the work are. What I want to say is that somehow the synthesis of the work has been led to the readers, instead of building the story of the paper in a direct and more clear way. I'm not saying that the theoretical results should not be included in the main manuscripts, but some of the proofs could be perhaps moved to the appendix and glue all technical details a bit better to understand where the contribution of each one is important for the final algorithm. In this way I'm referring to the sections 3.2, 3.3 and 3.4 mainly. On the other hand I think that the presentation of the algorithm is great and it seems to be a fabulous result (at least to me), but one has to dig quite a lot to find what it the role of the previous elements in that one. This is important, particularly for a submission of this type, for reproducibility and clarity -- which in the end leads to impact, of course.

**Point 2.** The second one is related to the lack of empirical results. As I said, even being a paper with relevant contributions and technical discoveries, I had the feeling once I finished to read it that I would have liked to see some empirical results. Mainly on the performance on the algorithm for different sets  of truncated samples, dimensionality and types of probability distributions. I understand the class of paper that this one is, mostly theoretical, and that once one proves such good estimates of parameters on exponential family distributions, which are super general, but to me it does not prevent the benefits that empirical results could produce on the clarity and quality of the manuscript.


**Questions:**

**Q1** -- Is there any sort of truncation effect on the samples that the given algorithm could not deal with properly?

**Q2** -- Is there any fundamental difference between $\rho$ and $p_\theta$ in the Equations below L138, L139 and the probability distribution mentioned in L230?

**Q3** -- In L122, the Cov[x,x] is described as a matrix, later for the Lemmas 3.2 and 3.3, the same covariance is upper and lower bounded using scalars. How this bound of a matrix with scalars work? Are we bounding all elements in the matrix?

---

> ### Author Rebuttal · Authors · 2023-08-09
>
> We thank the reviewer for the valuable feedback and for appreciating our results despite the weaknesses stated! We will use this space to address the main weaknesses and answer the questions you listed.
>
> ### **Weaknesses:**
>
> 1. _"First, even if I liked the idea, the utility, the algorithm and the results provided in a significant way, I had a hard time reading it and kind of building myself the answers to what the main contributions and strengths of the work are. What I want to say is that somehow the synthesis of the work has been led to the readers, instead of building the story of the paper in a direct and more clear way... In this way I'm referring to the sections 3.2, 3.3 and 3.4 mainly. On the other hand I think that the presentation of the algorithm is great and it seems to be a fabulous result (at least to me), but one has to dig quite a lot to find what it the role of the previous elements in that one. This is important, particularly for a submission of this type, for reproducibility and clarity -- which in the end leads to impact, of course."_
>
> **Answer:** We apologize if the presentation of the results were not clear. We can (as suggested in response to Review Dgrb and the one to all reviewers) instead think of moving some less essential technical lemmas to the Appendix and use the main content pages to write more intuition and explanations. We can also re-organize the sections: the strong convexity and smoothness parameters in 3.1 depend on $p_\theta(S)$. After this, perhaps we can introduce the results for section 3.3 (to bound $p_\theta(S)$), and then 3.2. We welcome any feedback here!
>
> 2. _"The second one is related to the lack of empirical results... I understand the class of paper that this one is, mostly theoretical, and that once one proves such good estimates of parameters on exponential family distributions, which are super general, but to me it does not prevent the benefits that empirical results could produce on the clarity and quality of the manuscript."_
>
> **Answer:** Thank you for your feedback! As we also mentioned in other responses, we did not include numerical demonstrations due to (1) the theoretical results already taking a lot of space to present (including the appendices), (2) it is not usually included in similar/related works. It is also our opinion that while simple simulated experiments may be less helpful for this paper than using the limited space for more intuition and exposition, real-world application may be beyond the scope of the current work.
>
> ### **Questions:**
>
> 1. _"Is there any sort of truncation effect on the samples that the given algorithm could not deal with properly?"_
>
> **Answer:** In general, the algorithm can deal with arbitrary truncations of the support of the data generating distribution as long as the truncation set has a non-zero measure. For instance, if the truncation set lies in some lower dimension with zero mass under the data generating distribution, this cannot be dealt with (although any zero-mass set cannot have a truncated density written in the form in L138). If we cannot even decide whether a point is inside the set or not efficiently, this would also be an issue.
>
> 2. _"Is there any fundamental difference between_ $\rho$ _and_ $p_\theta$ _in the Equations below L138, L139 and the probability distribution mentioned in L230?"_
>
> **Answer:** The distinction between $\rho$ and $p_\theta$ are just for differentiating any density $\rho$ and an exponential family distribution parameterized by $\theta$. Thus the result in L230 holds for any density, not necessarily even an exponential family distribution.
>
> 3. _"In L122, the Cov[x,x] is described as a matrix, later for the Lemmas 3.2 and 3.3, the same covariance is upper and lower bounded using scalars. How this bound of a matrix with scalars work? Are we bounding all elements in the matrix?"_
>
> **Answer:** Since the covariance matrix is PSD, we are using the Loewner order to express its eigenvalue bounds (notice that the constants are multiplied by the identity matrix so it is indeed a matrix). We should have made bold the ‘I’ for the identity matrix; after making this change, we hope the statement will be clearer.

---

> > ### Comment · Reviewer_P6iL · 2023-08-15
> > **After rebuttal comment**
> >
> > Thanks to the authors for the detailed response and the clarifications to my main concerns. Reading the rest of comments from reviewers, it seems that we are on a similar direction of thoughts. In particular, I would like to remark that I'm happy with the current state of the paper, and I hope that authors will update the manuscript according to what has been said if the paper finally moves forward. However, I agree with reviewer UJWe that it is important to provide numerical support for a theoretical paper. Also, the reasons included in the global comment are somehow limited to me, but I accept them. Thus, according to the feedback provided in the rebuttal and assuming that authors will finally add some additional numerical experiments in the supplementary (as it has been recently said), I raise my score to support the acceptance of the paper.

---

### Official Review · Reviewer_Dgrb · 2023-07-26

**Soundness:** 3 good
**Presentation:** 3 good
**Contribution:** 3 good
**Rating:** 7
**Confidence:** 1

**Summary:**

The paper proves the convergence of projected stochastic gradient descent (SGD) on the maximum-likelihood problem associated with estimating the parameters of an exponential family given truncated samples.
This result generalises previous work that shows similar results for Gaussian distributions to exponential families. To this end, the paper assumes bounded variance, log-concavity, sufficient statistics being polynomials, and bounded conditional expectation in the tails of the distributions.
These assumptions imply the convergence of SGD and, thereby, the existence of a polynomial-time algorithm for parameter estimation given truncated samples.


**Strengths:**

I am writing this review as an outsider to truncated statistics.

That said, even as an outsider, I find this submission well-organised and readable despite its technicality, which speaks for the clarity of the exposition. I feel a little less like an outsider after studying this work.

The results appear to be novel, and I have the impression that constructing an algorithm that estimates any (sufficiently well-behaved) exponential family from truncated samples (and not "just" a Gaussian; quotation marks because this kind of analysis requires non-trivial work even for Gaussian distributions, see [10]) is a significant result.


**Weaknesses:**

As mentioned above, I am not an expert in truncated statistics.

Even though, overall, I think this is a nice paper, I would like to comment on some potential weaknesses:

1. Without having studied reference [10] in full detail, it seems that the analysis in the submitted paper mirrors that in [10] rather closely. A similarly structured analysis can be fine, but a more specific contextualisation of Section 3 may be warranted. I would like to hear what the authors think about this.


2. Although I am fully aware that this paper is theoretical, reading about the efficiency of the proposed algorithm makes me want to see the algorithm "in action". Numerical demonstrations only appear to be optional in this line of work: some related works include them (e.g. [15, 31]), and some do not (e.g. [10, 12, 14]). But I think the present paper would appeal to a broader audience if it contained a few demonstrations of projected SGD for learning parameters of exponential families (maybe including, but not restricted to, Gaussians).


3. I found the appendices really helpful in understanding this work (some contain crucial information for proving some of the results). However, it did not help the readability of this manuscript, having to jump back and forth between the appendix and the main paper. Some parts must inevitably move to the appendix to condense such a comprehensive analysis into nine pages. But this makes me wonder whether a conference with a page limit of nine pages is the appropriate venue for this work. (For context: since related work of comparable volume (e.g. [14, 15]) has been published at Neurips, there seems to be a precedence for accepting this submission despite its length).


**Questions:**

Covered by "Weaknesses" above.

**Limitations:**

Limitations appear to be discussed wherever appropriate.

---

> ### Author Rebuttal · Authors · 2023-08-09
>
> Firstly, we thank the reviewer for the careful reading of our paper. We are very happy to hear that the reviewer found the submission well-organized and readable despite the technicality. We will use this space to address the main weaknesses mentioned in your review:
>
> 1. _"Without having studied reference [10] in full detail, it seems that the analysis in the submitted paper mirrors that in [10] rather closely. A similarly structured analysis can be fine, but a more specific contextualisation of Section 3 may be warranted. I would like to hear what the authors think about this."_
>
> **Answer:** Thank you for your comments! We had structured the analysis in this way because it is a common structure used in the related works, and we felt that familiarity in presentation may help with the readability of the results. We agree though that perhaps we can compare/contrast the parts of the analysis which differ from the Gaussian case much more clearly in the main content. Perhaps in the line with the remarks in Section 3.1, we can include more remarks in Sections 3.2 and 3.3, such as expanding on the comment made after Lemma 3.5 to contrast that in [10], the mean and covariance are analyzed separately whereas in our setting we consider them together (as $\theta$).
>
> 2. _"Although I am fully aware that this paper is theoretical, reading about the efficiency of the proposed algorithm makes me want to see the algorithm "in action". Numerical demonstrations only appear to be optional in this line of work: some related works include them (e.g. [15, 31]), and some do not (e.g. [10, 12, 14]). But I think the present paper would appeal to a broader audience if it contained a few demonstrations of projected SGD for learning parameters of exponential families (maybe including, but not restricted to, Gaussians)."_
>
> **Answer:** We did not include numerical demonstrations due to (1) the theoretical results already taking a lot of space to present (including the appendices) and as you mention (2) it is not usually included in similar/related works. We also believe that the current work is a stand-alone result of its own value, and that much more can be done (beyond the scope of this work) to make this algorithm perform much better in practice using domain knowledge. We agree that real-world application of our work would be very valuable, but may be more appropriate to include in a significant extension of the current work.
>
> 3. _"I found the appendices really helpful in understanding this work (some contain crucial information for proving some of the results). However, it did not help the readability of this manuscript, having to jump back and forth between the appendix and the main paper. Some parts must inevitably move to the appendix to condense such a comprehensive analysis into nine pages. But this makes me wonder whether a conference with a page limit of nine pages is the appropriate venue for this work. (For context: since related work of comparable volume (e.g. [14, 15]) has been published at Neurips, there seems to be a precedence for accepting this submission despite its length)."_
>
> **Answer:** Thank you for your careful reading and feedback! We also had a hard time with deciding how best to write the current paper so that the reader can get the main ideas but without eliding the necessary technical results. Perhaps we can even move some of the technical lemmas to the Appendix to improve readability, and include more intuition and exposition in their stead. We would like to hear any feedback on this suggestion (from all reviewers).

---

> > ### Comment · Reviewer_Dgrb · 2023-08-11
> >
> > Thank you for the thorough reply.
> >
> > Regarding 1. and 3.: I think a few remarks similar to your suggestions would be helpful. But since the other reviewers do not seem to raise these points, I leave it to the authors to decide whether to use the space for alternative improvements instead.
> >
> > Regarding 2.: Thanks for clarifying. I agree that a real-world simulation study would be too much to ask. I am unsure whether I agree with your two reasons for not including a simulation. However, I cannot predict how much we would learn from a small toy example (hidden in one of the appendices). And a comprehensive empirical evaluation seems to be out of scope.
> >
> > Altogether, I continue to believe that a numerical study would be helpful, but I also think the paper is in decent shape without it.
> > After reading the authors' rebuttal, I will keep my (already positive) score.

---

> > > ### Author Response · Authors · 2023-08-11
> > >
> > > Thank you very much for reading our response and for your quick follow up! We appreciate your balanced feedback, and if the other reviewers agree that it would be important to have some small numerical example, we will make the efforts to include it in the final paper. Thank you again!

---

### Author Rebuttal · Authors · 2023-08-09

# **Response to all reviewers**

Firstly, we want to sincerely thank all the reviewers for their time and careful feedback. We also want to thank all the reviewers for appreciating our results and recognising the significance of our work!

We start with addressing some common questions and concerns and then we will respond to the individual questions of the reviewers.

### **Difficulty following flow of technical results (Section 3), presentation.**
* We appreciate all of the feedback and suggestions to improve the presentation of the technical results. We are happy to move some technical lemmas to the Appendix and add more intuition and explanations in Sections 3.2 and 3.3, which we hope will improve the readability of the paper and it is a minor change to the submitted paper. We are also happy to take any other suggestions that the reviewers believe will help the readability of the paper without excluding important details.

### **Practical motivations and examples.**
* We are happy to include additional examples and clarifications below, which should also be a minor change provided that we have the space to do so.
* We’d like to highlight several practical examples with important societal implications which can fall under this kind of framework:
    * **Healthcare.** In many healthcare applications, the source of truncation in data can arise from patient deaths, prohibitive costs, and other variables. For example, one can estimate the mean health care cost for patients with cancer, given that some patients cannot completely follow up due to costs or death. Here, the truncation set would be the patient covariates and realized health care costs; things lying outside of the truncation set would be the patient covariates and unrealized health care costs (which we do not observe). It may be reasonable to have an oracle provided by an expert (who can determine how many more visits and costs a patient with particular covariates should have had).
    * **Discrimination.** In other settings with systematic bias, such as those considered by the Stanford Open Policing Project, we can consider the truncation set to be the realized outcomes across different groups of people. For instance, one could try to estimate search thresholds (probabilities) across different races given the realized data from traffic stops; here, the data lying outside of the truncation set could be considered the unrealized stops that would have occurred for each group if all standards for search are held equal. In this case, we may need to consider the oracle to be based on the given, realized data.
    * **Geographic boundaries.** In [31], the Chicago crime data was used to demonstrate the use of TruncSM because the city boundary was expressed as a polygon in $\mathbb{R}^2$. We can also use this data but deciding that a point is in the set or not is easier than describing the complicated boundary that one needs to calculate the weight functions in [31]. In addition we can handle more complicated cases that [31] cannot handle such as if the truncation set has holes or “cracks” in it as can be the case for complicated geographic boundaries (e.g., with islands, lakes) or political ones (e.g., result of gerrymandering).

* Regarding the membership oracle access to the set, we would like to clarify that in practice we may even know what the set S is (but which can be arbitrarily complicated or hard to describe mathematically), but we only need to be able to decide whether a point is inside the set or not as far as the algorithm is concerned. In contrast, e.g., [31] requires the set to be a Lipschitz domain and needs a description of its boundary to compute a weighting function. We would like to point out that integration and sampling are difficult even knowing S if it is very complex (e.g., not convex, not connected), and argue that membership oracle access to S is a weak requirement. Furthermore, the prior work [10] shows that without even membership oracle access to a set, recovery of parameters is (information theoretically) impossible from truncated data.

### **Numerical demonstration.**
We had decided not to include numerical demonstrations due to:
* The scope of the paper is to provide provably correct algorithms for estimating exponential families from truncated data.
* Our theoretical results already take a lot of space to present and draw a complete enough picture on the provable guarantees that we can get. We believe that adding simple simulated experiments that just certify the theoretical results would add less value than making use of the space to add more intuition and details on the theoretical result which is the goal of this paper.
* While a real-world application, on the contrary to simple simulated experiments, would add great value to our results, we believe that such practical implementation will need a careful application of our algorithm and many details will depend on the domain knowledge of each particular application. We hence believe that such real-world experiments is a very interesting future direction but it lies beyond the scope of the current work.
* Numerical demonstrations are not always included in the prior literature and related papers in this line of work (e.g., [10, 12, 14, 26, 38]).

---

### Author Response · Authors · 2023-08-15
**Small Numerical Demonstrations Completed**

We again thank all the reviewers for the helpful feedback! Since it has been brought up by the reviewers that the current work would still benefit substantially by including some numerical demonstration (even if we need to include it in the supplementary), we have also implemented our simple algorithm for "multivariate" exponential distributions (i.e., the density is $p_\theta(x) = -\theta \cdot \exp(\theta^\top x)$, where $\theta$ is a vector)).

We are of the understanding that we cannot upload any links or revisions at this time, so this notice may not be helpful to the reviewers. However, we hope that the reviewers trust that we have completed some basic numerical examples during the (limited) discussion period, and can have faith that we can complete more (and improve the examples quickly done during the discussion period) by the camera-ready deadline.

To be more specific, we have implemented our algorithm for 2-, 5-, and 10-dimensional exponential distributions. The truncation sets in the first two exclude the mean value in all directions, and the 10-dimensional one excludes them in some directions. We initialize our parameters however with the truncated means (recalling that $E[X] = -1/\theta$ to solve the non-truncated MLE solution), and run the projected SGD procedure with learning rate 0.01 for 1500 iterations (and 1500 samples). Whether in 2, 5, or 10 dimensions, all converge in 1500 steps with error (in L2 norm) at most 0.1. We hope to improve upon these examples and include them in the final version. Any other feedback for numerical examples is appreciated.

---

### Decision · Program_Chairs · 2023-09-21

**Decision:**

Accept (poster)

**Comment:**

The reviewers have reached a positive consensus on this novel work in extending results in the Gaussian setting to exponential families. They propose to use projected SGD and analyze the algorithm adequately, and the contributions are relevant to the NeurIPS readership. The authors should consider the reviewer recommendations, especially with respect to numerical examples, in preparing the next iteration of the manuscript